# Self-Consuming Generative Models Go MAD

**Sina Alemohammad,**[*][†] **Josue Casco-Rodriguez,**[*][†] **Lorenzo Luzi,**[†] **Ahmed Imtiaz Humayun,**[†]
**Hossein Babaei,**[†] **Daniel LeJeune,**[‡] **Ali Siahkoohi,**[§] **Richard G. Baraniuk**[†]

[†] Department of ECE, Rice University; [‡] Department of Statistics, Stanford University;
[§] Department of CMOR, Rice University

## Abstract

Seismic advances in generative AI algorithms for imagery, text, and other data types have led to the temptation to use AI-synthesized data to train next-generation models. Repeating this process creates an autophagous ("self-consuming") loop whose properties are poorly understood. We conduct a thorough analytical and empirical analysis using state-of-the-art generative image models of three families of autophagous loops that differ in how fixed or fresh real training data is available through the generations of training and whether the samples from previous-generation models have been biased to trade off data quality versus diversity. Our primary conclusion across all scenarios is that *without enough fresh real data in each generation of an autophagous loop, future generative models are doomed to have their quality (precision) or diversity (recall) progressively decrease.* We term this condition Model Autophagy Disorder (MAD), by analogy to mad cow disease, and show that appreciable MADness arises in just a few generations.

Generation $t = 1$     $t = 3$     $t = 5$     $t = 7$     $t = 9$

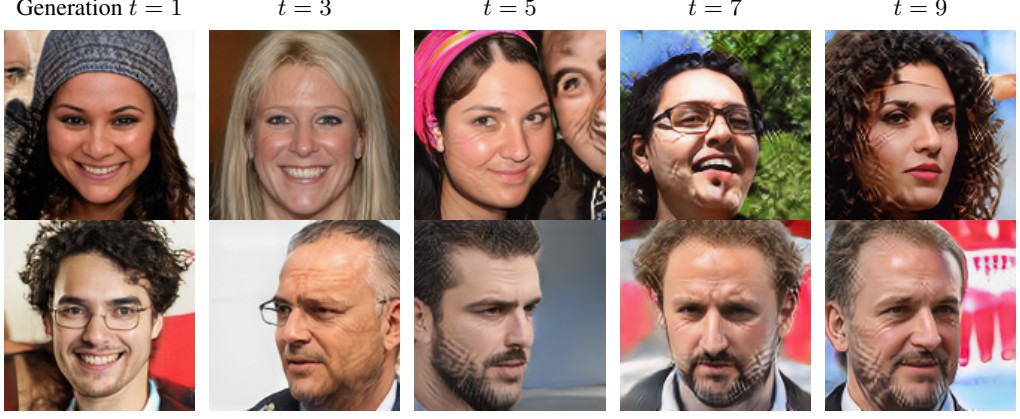

Figure 1: **Training generative artificial intelligence (AI) models on synthetic data progressively amplifies artifacts.** As AI-synthesized data proliferates in standard datasets and the Internet, future AI models will train on both real and synthetic data, forming *autophagous ("self-consuming") loops*. We highlight a potential unintended consequence of autophagous training. We trained a sequence of StyleGAN2 (Karras et al., 2019a) models wherein the model at generation $t \geq 2$ trains only on data synthesized by the model at generation $t - 1$. This forms a fully synthetic loop (Figure 3) without sampling bias ($\lambda = 1$). Note how the cross-hatched artifacts (possibly an architectural *fingerprint* (Karras et al., 2021)) are progressively amplified at each generation. Appendix D has more samples.

## 1 Introduction

Synthetic data[1] from *generative artificial intelligence (AI)* models like Stable Diffusion (Rombach et al., 2022) and ChatGPT (OpenAI, 2023) is rapidly proliferating on the Internet. Indeed, there will soon be much more synthetic data than real data on the Internet.

---

[*] Equal contribution.

[1] By "synthetic" we mean AI-synthesized data, as opposed to data synthesized via physics-based simulations.

Figure 2: **Today's large-scale image training datasets contain AI-generated data.** Datasets like LAION-5B (Schuhmann et al., 2022), which trains popular models like Stable Diffusion (Rombach et al., 2022), contain AI-synthesized images. Here are LAION-5B (haveibeentrained.com) samples containing data synthesized by (left to right) AICAN (Elgammal et al., 2017), Pix2Pix (Isola et al., 2017), StyleGAN (Karras et al., 2019a), and DALL-E (Ramesh et al., 2021). Generative models using LAION-5B thus close an autophagous ("self-consuming") loop (see Figure 3) that can progressively amplify artifacts (Figure 1) and lower quality (precision) and diversity (recall).

Since the training datasets for generative AI models tend to be sourced from the Internet, today's AI models are unwittingly being trained on increasing amounts of AI-synthesized data. Figure 2 confirms that the popular LAION-5B dataset (Schuhmann et al., 2022), used to train state-of-the-art models like Stable Diffusion, contains synthetic images from several earlier generations of generative models. AI is generating formerly human-sourced data, like reviews (Gault, 2023), websites (Cantor, 2023), and data annotations (Veselovsky et al., 2023), often with no indication of its origin (Christian, 2023). As the use of generative models continues to grow rapidly, this situation will only accelerate.

Moreover, throwing caution to the wind, AI-synthesized data is increasingly used by choice for training because it is convenient, especially in data-scarce applications like medicine (Pinaya et al., 2022) and geophysics (Deng et al., 2022), and because it can protect privacy (Luzi et al., 2024; Klemp et al., 2023) in sensitive data applications like medicine (Packhäuser et al., 2022; DuMont Schütte et al., 2021). Most importantly, as deep learning models become increasingly enormous (Azizi et al., 2023; Burg et al., 2023), we are simply running out of real data on which to train them (Economist, 2023a;b; Villalobos et al., 2022).

The witting or unwitting use of synthetic data to train generative models departs from standard AI training practice in one important respect: repeating this process for generation after generation of models forms an **autophagous ("self-consuming") loop** (Figure 3). Different autophagous loop variations arise depending on how existing real and synthetic data are combined into future training sets. Additional variations arise depending on the model *sampling biases* used to trade off perceptual *quality* (fidelity or coherence) versus *diversity* (variety or heterogeneity).[2]

The potential ramifications of autophagous loops on the properties and performance of generative models is poorly understood. In one direction, autophagy might progressively amplify the biases and artifacts present in any generative model as *fingerprints*. In Figure 1, autophagy progressively amplifies cross-hatching artifacts (reminiscent of aliasing (Karras et al., 2021)) in subsequent generations of StyleGAN2 models. In another direction, autophagous loops featuring generative models tuned to produce high quality syntheses at the expense of diversity (such as Karras et al. (2019a); Ho and Salimans (2021)) might progressively dilute the diversity of the data on the Internet.[3] By analogy to *mad cow disease* (Nathanson et al., 1997), we term these and other symptoms of autophagy as *Model Autophagy Disorder (MAD)*.

**Contributions.** We conduct a careful theoretical and empirical study of AI autophagy with generative image models. The concepts developed herein apply to any data type, including text. We also unify the results of contemporaneous work. Our three key contributions establish that, *without enough fresh real data each generation, future generative models are doomed to go MAD.* Moreover, we demonstrate that appreciable MADness can occur in only a handful of generations.

**1. Realistic models for autophagous loops.** We propose three families of self-consuming loops that realistically model how real and synthetic data are used to train generative models (recall Figure 3):

---

[2]We quantify quality and diversity via *precision* and *recall*, respectively (Kynkäänniemi et al., 2019).

[3]Similar to *diversity exposure* in recommender systems, where maximizing click rates discourages exposure to diverse ideas (Stroud, 2011; Dylko et al., 2017; Beam, 2014; Bakshy et al., 2015; O'Callaghan et al., 2015).

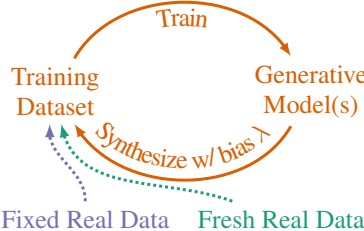

Figure 3: **Recursively training generative models on synthetic data from other models produces an autophagous ("self-consuming") loop.** In this paper, we study three autophagous loop variants (defined in Section 2): the *fully synthetic loop* (only synthetic data), the *synthetic augmentation loop* (synthetic + fixed real data), and the *fresh data loop* (synthetic + fresh real data). Each generation samples with a bias $\lambda$ that trades off sample quality versus diversity.

- The fully synthetic loop (Section 3), where the training dataset for each generation's model consists solely of synthetic data sampled from previous generations' models, such as by training a generative model on its own outputs (followfox.ai, 2023). We show that *either the quality (precision) or the diversity (recall) of the generative models decreases over generations.*

- The synthetic augmentation loop (Section 4), where each generation's training data includes syntheses from previous generations and a fixed set of real data, such as by training on real and self-generated data (Huang et al., 2022). We show that *fixed real training data only delays the inevitable degradation of the quality or diversity of the generative models over generations.*

- The fresh data loop (Section 5), where each generation's training data includes syntheses from previous generations plus some fresh real data, which models the common practice of scraping the Internet for training data, which finds both real and synthetic data (recall Figure 2). We are the first to propose and study this flavor of autophagy. We show that, *with enough fresh real data, model convergence is independent of initialization, with quality and diversity that do not degrade over generations.*

**2. Sampling bias plays a key rôle in autophagous loops.** Practitioners often favor high-quality syntheses, whether through curation or automatic quality-diversity tradeoffs (Ho and Salimans, 2021; Karras et al., 2020). We show that, without these *sampling biases*, MADness degrades quality and diversity, while with them, quality can be maintained but diversity degrades even faster.

**3. Autophagous loop behaviors hold across a wide range of generative models and datasets**, including Gaussian, Gaussian mixture, diffusion (DDPM, Ho et al., 2020), StyleGAN2 (Karras et al., 2020), Wasserstein GAN (WGAN, Gulrajani et al., 2017a), and Normalizing Flow (Kobyzev et al., 2020) models trained on datasets like FFHQ (Karras et al., 2019b) and MNIST (Deng, 2012).

**Related work.** We define a cohesive autophagy framework, supported empirically by state-of-the-art models, that unifies and significantly extends contemporaneous results that consider fragmented aspects of MADness. Shumailov et al. (2023) study autophagous loops without sampling bias and show that MADness ensues from variational autoencoders and Gaussian mixture models in fully synthetic and language models in synthetic augmentation loops. However, the absence of quality-diversity tradeoffs (sampling biases) in their models limits the applicability of their findings to real-world scenarios. Furthermore, in each generation they only fine-tune their language models, while we train our models from scratch. Martínez et al. (2023a) also consider unbiased synthetic augmentation loops, but only show qualitative evidence of MADness on a small dataset. Martínez et al. (2023b) focus only on fully synthetic loops and report that sampling bias can prevent degradation of image quality in small datasets. Finally, Huang et al. (2022); Hataya et al. (2022) and others have considered synthetic data augmentation, but not in the context of autophagous loops.

## 2 SELF-CONSUMING GENERATIVE MODELS

Consider a sequence of generative models $(\mathcal{G}^t)_{t\in\mathbb{N}}$, where each model approximates a reference probability distribution $\mathcal{P}_r$. At each *generation* $t \in \mathbb{N}$, the model $\mathcal{G}^t$ trains from scratch on the dataset $\mathcal{D}^t = (\mathcal{D}_r^t, \mathcal{D}_s^t)$ containing both $n_r^t$ *real samples* $\mathcal{D}_r^t$ from $\mathcal{P}_r$ and $n_s^t$ *synthetic samples* $\mathcal{D}_s^t$ from trained generative model(s). The first-generation model $\mathcal{G}^1$ trains only on real data: $n_s^1 = 0, \mathcal{D}_s^1 = \emptyset$.

**Definition.** An *autophagous generative process* is a sequence of distributions $(\mathcal{G}^t)_{t\in\mathbb{N}}$ where each generative model $\mathcal{G}^t$ is trained on data that includes samples from previous models $(\mathcal{G}^\tau)_{\tau=1}^{t-1}$.

**Definition.** Let $\mathrm{dist}(\cdot, \cdot)$ denote a distance metric on distributions. A *MAD generative process* is a sequence of distributions $(\mathcal{G}^t)_{t\in\mathbb{N}}$ such that $\mathbb{E}[\mathrm{dist}(\mathcal{G}^t, \mathcal{P}_r)]$ grows with $t$.

**Claim.** *Under mild conditions, an autophagous generative process is a MAD generative process.*

Two critical aspects affect whether a sequence of generative models goes MAD: the balance of real and synthetic training data and how the generative models synthesize data. We study three realistic autophagous mechanisms, each of which includes synthetic data and potentially real data in a feedback loop (recall Section 1 and Figure 3):

- **The fully synthetic loop:** Each model $\mathcal{G}^t$ for $t \geq 2$ trains exclusively on synthetic data sampled from models $(\mathcal{G}^\tau)_{\tau=1}^{t-1}$ from previous generations, i.e., $\mathcal{D}^t = \mathcal{D}_s^t$.

- **The synthetic augmentation loop:** Each model $\mathcal{G}^t$ for $t \geq 2$ trains on a dataset $\mathcal{D}^t = (\mathcal{D}_r, \mathcal{D}_s^t)$: a fixed set of real data $\mathcal{D}_r$ from $\mathcal{P}_r$, plus synthetic data $\mathcal{D}_s^t$ from previous generations' models.

- **The fresh data loop:** Each model $\mathcal{G}^t$ for $t \geq 2$ trains on a dataset $\mathcal{D}^t = (\mathcal{D}_r^t, \mathcal{D}_s^t)$: a fresh set of real data $\mathcal{D}_r^t$ drawn from $\mathcal{P}_r$, plus synthetic data $\mathcal{D}_s^t$ from previous generations' models.

**Metrics for MADness.** Throughout this paper we measure the distance between the synthetic data and the real data (reference distribution) using the Fréchet inception distance (FID) (Heusel et al., 2017),[4] the quality of the synthetic data using *precision*, and the diversity of the synthetic data using *recall* (Kynkäänniemi et al., 2019). See Appendix A.4 for more details.

## 2.1 BIASED SAMPLING IN AUTOPHAGOUS LOOPS

While the above three autophagous loops realistically mimic real-world generative model training scenarios that involve synthetic data, it is also critical to consider how each generation's synthetic data is produced in practice. In particular, most syntheses are to some degree biased to maximize perceptual quality, whether through manual curation ("cherry-picking") or common techniques that automatically boost quality and sacrifice diversity by sampling closer to the modes of the synthetic distribution of the generative model (OpenAI, 2023; Ho and Salimans, 2021; Karras et al., 2020; Brock et al., 2019; Humayun et al., 2022). We refer to this common practice as *sampling bias*. We employ a number of generative models in our experiments below; each has a unique controllable parameter to increase sample quality. We unify these parameters in the universal *sampling bias parameter* $\lambda \in [0, 1]$, where $\lambda = 1$ corresponds to unbiased sampling and $\lambda = 0$ corresponds to sampling from the modes of the generative distribution $\mathcal{G}^t$ with zero variance. The exact interpretation of $\lambda$ differs across various models, but in general synthetic sample quality will increase and diversity will decrease as $\lambda$ is decreased from 1. Below we provide specific definitions for $\lambda$ for the various generative models we consider in this paper:

**Gaussian models:** To implement biased sampling from an estimated distribution $\mathcal{N}(\boldsymbol{\mu}, \boldsymbol{\Sigma})$, we sample from $\mathcal{N}(\boldsymbol{\mu}, \lambda\boldsymbol{\Sigma})$. As $\lambda$ decreases, we draw samples closer to the mean $\boldsymbol{\mu}_t$.

**Generative adversarial networks:** In our StyleGAN2 experiments, we decrease the truncation parameter $\Psi \in [0, 1]$ to increase sample quality (Karras et al., 2020). Thus, $\lambda = \Psi$.

**Diffusion models:** For DDPMs, we use a classifier-free diffusion guidance factor $w$ (with $10\%$ conditioning dropout) (Ho and Salimans, 2021) and define $\lambda = \frac{1}{1+w}$.

## 3 THE FULLY SYNTHETIC LOOP: TRAINING EXCLUSIVELY ON SYNTHETIC DATA LEADS TO MADNESS

First, we analyze the fully synthetic loop, where each model trains on synthetic data from previous generations. We focus on the inter-generational propagation of non-idealities resulting from estimation errors and sampling biases and characterize the convergence of the autophagous loop. The fully synthetic loop's simplicity primarily reflects niche examples like training generative models on their own high-quality outputs (followfox.ai, 2023). Nevertheless, this loop represents a worst-case scenario that provides insights into the more practical autophagous loops discussed in subsequent sections. Our analysis and experiments support our main conclusion for the fully synthetic loop: *either the quality (precision) or the diversity (recall) of the synthetic data deteriorates over generations.*

---

[4]We calculate MNIST FIDs via LeNet (Lecun et al., 1998) features instead of Inception features.

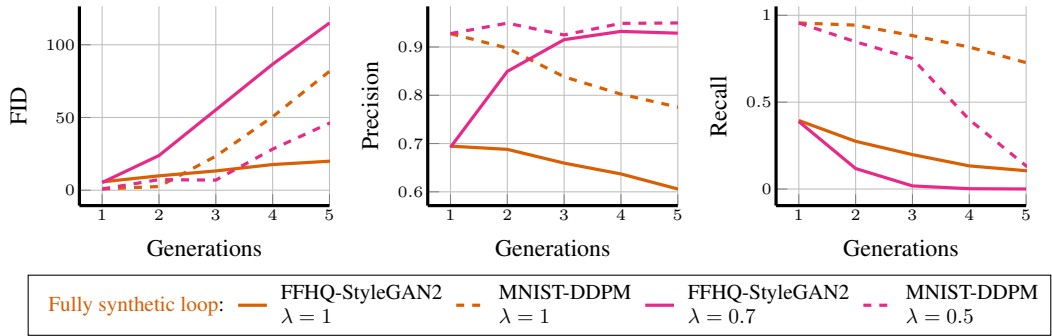

Figure 4: **Training generative models in a fully synthetic loop reduces both the quality and diversity of synthetic data, depending on sampling bias.** We plot the FID, precision (quality), and recall (diversity) of the synthetic FFHQ and MNIST images from a fully synthetic loop with unbiased ($\lambda = 1$) and biased ($\lambda < 1$) StyleGAN2 and DDPM models. See Figure 1 for StyleGAN2 samples demonstrating that the fully synthetic loop amplifies sample artifacts. In all cases, FID increases and diversity decreases. However, sampling bias can salvage quality (at the expense of diversity).

**Gaussian fully synthetic loop: random walks and variance collapse.** We first show that these loops have a martingale nature that causes MADness. Consider a reference distribution $\mathcal{P}_r = \mathcal{N}(\boldsymbol{\mu}_0, \boldsymbol{\Sigma}_0)$, where $\boldsymbol{\mu}_0 \in \mathbb{R}^d$ and $\boldsymbol{\Sigma}_0 \in \mathbb{R}^{d \times d}$, and a Gaussian generative process $\mathcal{G}^t = \mathcal{N}(\boldsymbol{\mu}_t, \boldsymbol{\Sigma}_t)$. At each time $t \in \mathbb{N}$, we sample $n_s$ vectors from $\mathcal{G}^{t-1}$ with bias $\lambda \leq 1$; i.e., we draw $\mathbf{x}_t^1, \ldots, \mathbf{x}_t^{n_s} \overset{\text{iid}}{\sim} \mathcal{N}(\boldsymbol{\mu}_{t-1}, \lambda \boldsymbol{\Sigma}_{t-1})$. From these vectors we construct the unbiased parameters of the next model $\mathcal{G}^t$:

$$\boldsymbol{\mu}_t = \frac{1}{n_s} \sum_{i=1}^{n_s} \mathbf{x}_t^i, \qquad \boldsymbol{\Sigma}_t = \frac{1}{n_s - 1} \sum_{i=1}^{n_s} (\mathbf{x}_t^i - \boldsymbol{\mu}_t)(\mathbf{x}_t^i - \boldsymbol{\mu}_t)^\top. \tag{1}$$

It is straightforward to see that $\boldsymbol{\mu}_t$ and $\boldsymbol{\Sigma}_t$ are (super)martingale processes (Williams, 1991) that take random walks. For $\boldsymbol{\Sigma}_t$, we also have the following result that is proved in Appendix B.

**Proposition.** *For the random process defined in Equation (1), for any $\lambda \leq 1$, we have $\boldsymbol{\Sigma}_t \xrightarrow{\text{a.s.}} \mathbf{0}$.*

That is, when we repeatedly fit a distribution to data sampled from that distribution, we should not only expect some modal drift because of the random walk in $\boldsymbol{\mu}_t$ (reduction in *quality*) but also inevitably a collapse of the variance $\boldsymbol{\Sigma}_t$ (vanishing of *diversity*).

The key takeaway is that these effects—the random walk and the variance collapse—are solely due to the estimation error of fitting the model parameters using random data. Importantly, this result holds true even when there is no sampling bias ($\lambda = 1$). The magnitudes of the steps of the random walk in $\boldsymbol{\mu}_t$ are determined by two main factors: the number of samples $n_s$ and the covariance $\boldsymbol{\Sigma}_t$. Unsurprisingly, the larger the $n_s$, the smaller the steps of the random walk, since there will be less estimation error. This will also slow the convergence of $\boldsymbol{\Sigma}_t$ to $\mathbf{0}$. Meanwhile, $\boldsymbol{\Sigma}_t$ can be controlled using a sampling bias factor $\lambda < 1$. The smaller the choice of $\lambda$, the more rapidly $\boldsymbol{\Sigma}_t$ will converge to $\mathbf{0}$, stopping the random walk of $\boldsymbol{\mu}_t$ (as illustrated in Figure 15). Thus, the sampling bias factor $\lambda$ provides a trade-off to preserve quality at the expense of diversity. Shumailov et al. (2023) recently showed that the expected Wasserstein-2 distributional distance $\mathbb{E}[\text{dist}(\mathcal{G}^t, \mathcal{P}_r)]$ increases in this process, supporting our conclusion that $\mathcal{G}^t$ is a MAD generative process.

We now empirically study the fully synthetic loop using FFHQ-trained StyleGAN2 and MNIST-trained DDPM models; see Appendix A.1 for the experimental details.

**Unbiased sampling degrades synthetic data quality and diversity.** Figure 4 plots the FID, precision, and recall for FFHQ-StyleGAN2 and MNIST-DDPM models in fully synthetic loops with ($\lambda < 1$) and without ($\lambda = 1$) sampling bias. In the latter case, the synthetic data distributions undergo random walks that deviate from the reference distribution because each generation's training data is finite. Consequently, the models go MAD: FID increases, while precision and recall steadily decrease.

**Biased sampling can boost synthetic data quality, but at the expense of diversity.** As for the biased FFHQ-StyleGAN2 and MNIST-DDPM models ($\lambda = 0.7$ and $0.5$) in fully synthetic loops

Generation $t = 1$  $\qquad$ $t = 3$  $\qquad$ $t = 5$

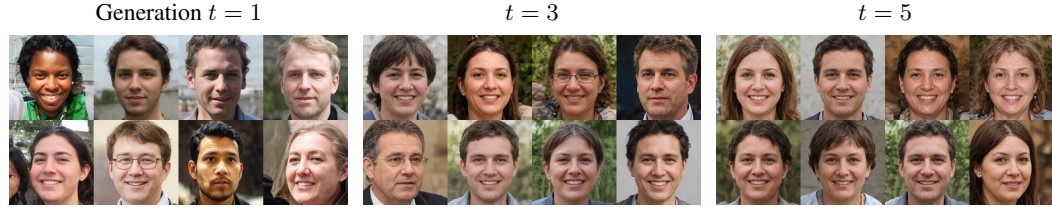

Figure 5: **Training generative models on biased synthetic data in a fully synthetic loop progressively loses diversity.** We repeat the experiment from Figures 1 and 4 but with sampling bias $\lambda = 0.7$. The randomly selected syntheses clearly lose diversity. Appendix E displays additional samples.

Generation $t = 3$  $\xrightarrow{\lambda = 1}$  $t = 10$  $\qquad$ Generation $t = 3$  $\xrightarrow{\lambda = 0.8}$  $t = 10$

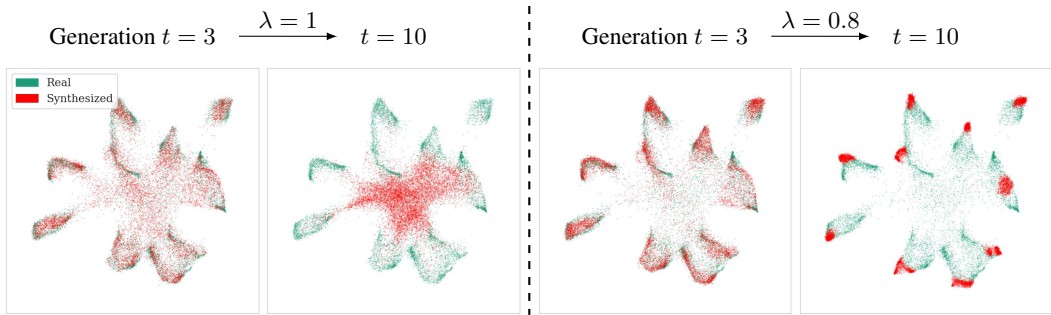

Figure 6: **In the fully synthetic loop, unbiased sampling loses quality, while biased sampling loses diversity.** 2D UMAP projections of $784$-dimensional MNIST-DDPM samples from fully synthetic loops without (left, $\lambda = 1$) and with (right, $\lambda < 1$) sampling bias. Without sampling bias, the synthetic digits become so unrealistic (low-quality) that they are easily distinguishable from real digits, while with sampling bias, the digits remain realistic but progressively lose diversity. See Appendix F for the synthesized samples.

(see Figure 4), sampling bias increases precision but also accelerates losses in recall (shown clearly in Figure 5) compared to unbiased models. Moreover, the FID still increases, indicating a MAD generative process. See Figure 14 in Appendix C.3 for results with different MNIST-DDPM sampling bias values, which follow the same trend.

**Synthetic mode behavior depends on the sampling bias.** To visualize MAD generative processes, in Figure 6 we reduced the dimensionality of the real and synthetic MNIST-DDPM fully synthetic loop samples from Figure 4 via Uniform Manifold Approximation and Projection (UMAP) (McInnes et al., 2018). With unbiased sampling, the ten modes of the synthetic distribution (one for each digit) progressively drift away from the real distribution modes, despite originating from a conditional model, and eventually merge into one large cluster. By generation $t = 10$, the synthetic digits are illegible (Figure 24 in Appendix F). In sharp contrast to the unbiased case, UMAP reveals that biased sampling successfully keeps syntheses on the real data manifold (high precision), but contracts the synthetic support around a single set of ten digits (zero recall). Appendix C confirms these trends for Gaussian mixtures, WGANs, and Normalizing Flows.

## 4 THE SYNTHETIC AUGMENTATION LOOP: FIXED REAL TRAINING DATA CAN DELAY BUT NOT PREVENT MADNESS

While analysis of the fully synthetic loop is straightforward, practitioners will use real data when it is available. We now explore the synthetic augmentation loop, where a fixed real dataset is augmented with autophagous synthetic data. Synthetic data augmentation can improve classification (Luzi et al., 2024; Burg et al., 2023), but the impact of autophagous data augmentation is unclear—does increasing training data volume enhance synthesis, even if the added samples stray from reality? We find that, in the synthetic augmentation loop, *fixed real training data only delays the inevitable degradation of the quality or diversity of synthetic data over generations.*

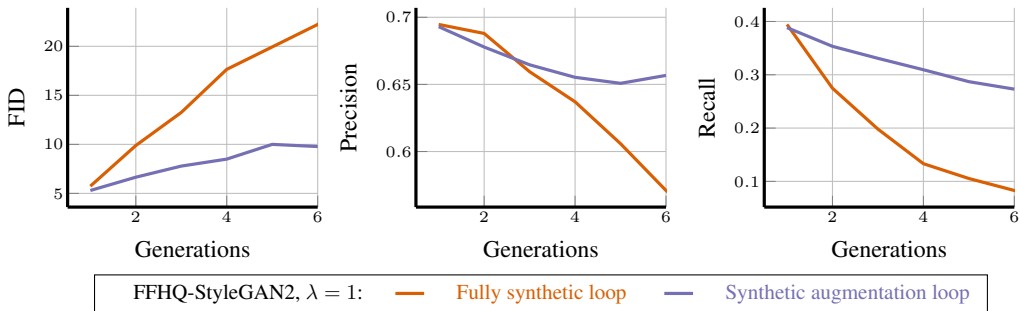

Figure 7: **Training generative models in a synthetic augmentation loop without sampling bias reduces synthetic quality and diversity, albeit more slowly than in the fully synthetic loop.** We plot the FID, precision (quality), and recall (diversity) of FFHQ-StyleGAN2 syntheses from a synthetic augmentation loop, wherein generative models train on both synthetic and real data, and a fully synthetic loop from Figure 4 for comparison. Both loops have no sampling bias ($\lambda = 1$). Qualitative examples (Appendix G) show the same artifacts as in Figure 1, albeit less prominently.

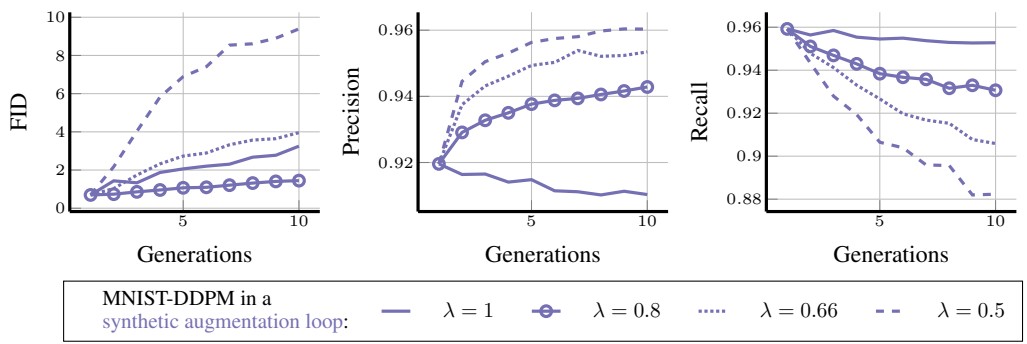

Figure 8: **When incorporating real data in the synthetic augmentation loop, sampling bias still affects MADness.** We plot the FID, precision (quality), and recall (diversity) of MNIST-DDPM images synthesized in synthetic augmentation loops with different sampling biases $\lambda$. All three metrics exhibit the same, albeit less pronounced, behavior as in the biased fully synthetic loops depicted in Figure 4.

**Keeping the original real dataset in the synthetic augmentation loop only slows MADness.** Figure 7 shows that keeping the full FFHQ dataset in a StyleGAN2[5] synthetic augmentation loop still produces the same symptoms (albeit more slowly) as the fully synthetic loop: the distance from the real dataset (FID) increases, while the quality (precision) and diversity (recall) of synthetic samples still decrease without sampling bias. (See Appendix A.2 for the experimental details.) In fact, in Appendix G we see the same artifacts as in Figure 1 and Appendix D. Additionally, sampling bias $\lambda$ impacts MNIST-DDPM synthetic augmentation loops (Figure 8) in the same way it impacts fully synthetic loops: FID still increases, but $\lambda < 1$ can increase quality (precision) in exchange for diversity (recall). Additional synthetic augmentation loop experiments can be found in Appendix H.

## 5 THE FRESH DATA LOOP: FRESH REAL DATA CAN PREVENT MADNESS

Our most elaborate autophagous loop model obtains training data from two sources: unseen (fresh) real data and synthetic data from previously trained models. A clear instance of this is the LAION-5B dataset (Schuhmann et al., 2022), which contains both real and AI-synthesized images from the Internet (Figure 2). We seek to understand how generative models evolve in the fresh data loop, which alters the synthetic augmentation loop by incorporating fresh (instead of fixed) real samples at

---

[5]Unique to our StyleGAN2 synthetic augmentation loop, we linearly grow a pool of synthetic data to assess whether access to all previous generations' synthetic data could help future generations learn (see Appendix A.2).

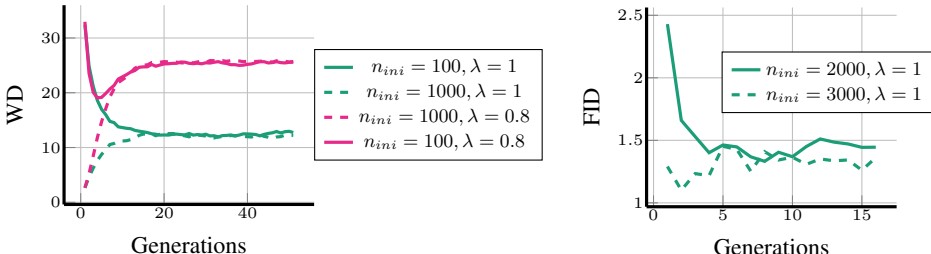

Figure 9: **In a fresh data loop, generative models converge to a state independent of the initial generative model**. We plot the Wasserstein distance (WD) and FID of two fresh data loop models: a Gaussian with $n_r = 100, n_s = 900$ (left) and an MNIST-DDPM with $n_r = n_s = 2000$ (right). We simulate the former with both unbiased and biased sampling. Across all models, we see that the asymptotic WD and FID are independent of the initial real samples $n_{ini}$ and the initial WD or FID.

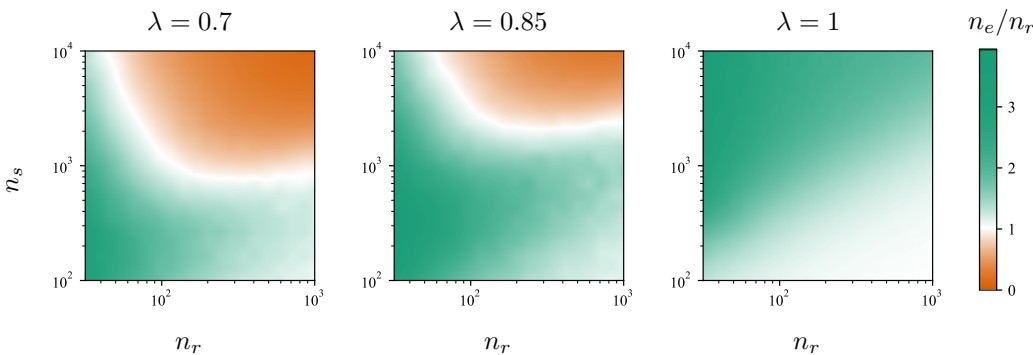

Figure 10: **In a fresh data loop, the admissible amount of synthetic data does not increase with the amount of real data.** As the real data count $n_r$ increases, the synthetic data count $n_s$ for which $n_e \geq n_r$ (**green** area) converges. Synthetic data is only likely to be helpful for small $n_r$.

each iteration. We imagine that a fraction $p \in (0, 1)$ of a corpus of data (e.g., the Internet) is real, and the remainder $1 - p$ is synthetic. Independently sampling $n^t$ data points from this corpus yields $n_r^t = pn^t$ real and $n_s^t = (1 - p)n^t$ synthetic data points to train the $t$-th generation model.

The fresh data loop reveals two intriguing phenomena. First, asymptotic performance converges independently of initial performance, depending only on the ratio of real-to-synthetic training data. Second, limited amounts of synthetic data can actually improve performance in the fresh data loop—since synthetic data propagates information from previously seen real data and thus increases the effective dataset size—but too much synthetic data can still cause MADness. Overall, our fresh data loop analyses and experiments establish that, *with enough fresh real data, the quality and diversity of synthetic data do not degrade over generations.*

**Initial models will eventually be forgotten in the fresh data loop.** First we show that the initial model does not affect the behavior of the fresh data loop. We train the initial model on $n_{ini}$ real samples and subsequent models with $n_r$ new real and $n_s$ synthetic (with bias $\lambda$) samples from the previous model; see Appendix A.3 for the details. Interestingly, for both the Gaussian and MNIST-DDPM and models, the Wasserstein distance and FID converge independently of $n_{ini}$ after a few iterations (Figure 9). In other words, we observe that models in a fresh data loop converge to a limit point that depends on $n_r, n_s$, and $\lambda$, but not on the initial model $\mathcal{G}^1$ or its dataset size $n_{ini}$:

$$\lim_{t\to\infty} \mathbb{E}[\text{dist}(\mathcal{G}^t, \mathcal{P}_r)] =: \text{WD}(n_r, n_s, \lambda). \tag{2}$$

For autophagy, this brings some hope: with fresh real data at each generation, $\mathbb{E}[\text{dist}(\mathcal{G}^t, \mathcal{P}_r)]$ does not necessarily increase with $t$. *In other words, a fresh data loop does not necessarily go MAD.*

**The fresh data loop exhibits a phase transition.** Since fresh real data can mitigate MADness, one might suspect that synthetic data can provoke fresh data loop MADness. However, the truth is that modest amounts of synthetic data in fresh data loops can actually *boost performance*; only when the

amount of synthetic data exceeds a critical threshold do models suffer. We formalize this observation through Monte-Carlo simulation of the fresh data loop limit point (Equation (2)) in Gaussian models. For comparison, we compute the *effective sample size* $n_e$ that an alternative model would need to reach the same performance as the asymptote from scratch:

$$\text{Find} \quad n_e \quad \text{s.t.} \quad \mathbb{E}[\text{dist}(\mathcal{G}(n_e), \mathcal{P}_r)] = \text{WD}(n_r, n_s, \lambda). \tag{3}$$

That is, $n_e$ captures the asymptotic sample efficiency of the fresh data loop.

Figure 10 depicts how the ratio $n_e/n_r$ changes with $n_r, n_s$, and $\lambda$. When $n_e/n_r \geq 1$, we say the amount of synthetic data $n_s$ is *admissible* because it effectively increases the number of real samples. For $n_e/n_r < 1$, synthetic data effectively reduces the number of real samples.

First, we confirm that, given $n_r$ and $\lambda < 1$, there exists a phase transition in $n_s$. If $n_s$ exceeds the admissibility threshold, then the effective sample size $n_e$ drops below the fresh sample size $n_r$, meaning that synthetic data does not asymptotically improve performance. However, the synthetic-to-real ratio $n_s/n_r$ needed to achieve $n_e/n_r \geq 1$ is not constant. In fact, in Figure 10 we see that the admissible amount of synthetic data $n_s$ (such that $n_e/n_r \geq 1$) can be quite high for small values of $n_r$, but as $n_r$ grows, the admissible ratio of synthetic-to-real data $n_s/n_r$ shrinks.

Second, we find that the admissible threshold value for $n_s$ depends strongly on the sampling bias $\lambda$. Perhaps surprisingly, stronger bias (smaller $\lambda$) actually reduces the number of synthetic samples that can be used without harming performance. Taking the limit $\lambda \to 1$ for unbiased sampling appears to ensure that the effective number of samples is always increased ($n_e/n_r$ is always greater than 1). Whether this limiting behavior extends beyond Gaussian models is an open question. As we discussed in Section 2.1, it is unlikely that practical generative models synthesize without bias, and so it is better to draw conclusions from the $\lambda < 1$ case. See Appendix I for additional fresh data loop experiments.

## 6 DISCUSSION

Our theoretical and empirical analyses have enabled us to extrapolate what might happen as generative models become ubiquitous and train future models in autophagous (self-consuming) loops. Using state-of-the-art generative image models and datasets, we have studied three families of autophagous loops and identified the key rôle of sampling bias. Some ramifications are clear: without enough fresh real data, future generative models are doomed to Model Autophagy Disorder (MAD), progressively losing quality (precision) or diversity (recall) and amplifying generative artifacts. One doomsday scenario is that, if left uncontrolled, MAD could poison the entire Internet's data quality and diversity. After all, our autophagous loops went appreciably MAD after just 5 generations (Figure 1). It seems inevitable that AI autophagy's unintended consequences could arise in the near future.

Practitioners who deliberately use synthetic training data should heed our warning. For those in truly data-scarce applications, our results suggest how much real data can prevent MADness. For example, future training of a medical image generator on inter-institutional anonymous syntheses (DuMont Schütte et al., 2021) should ensure that all synthetic images are artifact-free and diverse (see Section 3), and that real (preferably new) data is maximally present in training (see Sections 4 and 5).

Practitioners who unknowingly train on synthetic data could try controlling the ratio of real-to-synthetic training data by identifying and rejecting synthetic data. Some identifiers find telltale patterns of AI synthesis (Guarnera et al., 2020; Mitchell et al., 2023; Tang et al., 2023). Others make synthetic data steganographically idenfiable via *watermarking* (Kirchenbauer et al., 2023a;b; Zhao et al., 2023; Peng et al., 2023; Wen et al., 2023; Fernandez et al., 2023; Fei et al., 2022). However, watermarking deliberately introduces hidden artifacts that could be uncontrollably or harmfully amplified by autophagy. In the fresh data loop, modest amounts of synthetic data can boost performance ($n_e/n_r > 1$ in Figure 10). Future research could develop *autophagy-aware watermarking* that helps identify synthetic data while avoiding the amplification of its own artifacts.

Future research directions include combining our three prototypical autophagous loops into more complex loops, examining how MADness affects downstream tasks (e.g., classification), and models for other data types. We have focused here on imagery, but autophagy and MADness can occur in any data type. For example, autophagous language models (Huang et al., 2022; Wang et al., 2022; Taori et al., 2023) can also go MAD, losing quality (coherence or correctness) or diversity (variety). Shumailov et al. (2023) have reached similar conclusions, but there is much work to do in this vein.

## ACKNOWLEDGEMENTS

Thanks to H. Javadi, B. Mason, and S. Sonkar for their insights. This work was supported by NSF grants CCF-1911094, IIS-1838177, and IIS-1730574; ONR grants N00014-18-12571, N00014-20-1-2534, and MURI N00014-20-1-2787; AFOSR grant FA9550-22-1-0060; DOE grant DE-SC0020345; and a Vannevar Bush Faculty Fellowship, ONR grant N00014-18-1-2047. DL was supported by ARO grant 2003514594.

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

# A  EXPERIMENT SETUPS

Here are detailed descriptions of our experiments.

## A.1  THE FULLY SYNTHETIC LOOP

We empirically study the fully synthetic loop using two representative deep generative models and two practical training datasets. After training an initial model $\mathcal{G}^1$ with a fully real dataset containing $n_r^1$ samples from the (unknown) reference distribution, subsequent models $(\mathcal{G}^t)_{t=2}^{\infty}$ are trained using $n_s^t$ synthetic samples from the immediately preceding model $\mathcal{G}^{t-1}$, where each synthetic sample is produced with sampling bias $\lambda$. Our primary experiments are organized as follows:

- **Generative adversarial network:** We use an unconditional StyleGAN2 model (Karras et al., 2020) and initially train it on $n_r^1 = 70k$ samples from the FFHQ dataset (Karras et al., 2019b). We downsized the FFHQ images to $128 \times 128$ (using Lanczos PyTorch anti-aliasing filtering as in Karras et al. (2020)) to reduce the computational cost. We set $n_s^t = 70k$ for $t \geq 2$.

- **Diffusion model:** We use a conditional DDPM (Ho et al., 2020) with $T = 500$ diffusion time steps and initially train it on $n_r^1 = 60k$ real samples from the MNIST dataset. We set $n_s^t = 60k$ for $t \geq 2$. To calculate FIDs, we use the features extracted by a LeNet (Lecun et al., 1998) rather than an Inception network, because numerical digits are not exactly natural images. For consistency, we continue to use the term "FID" in this case.

## A.2  THE SYNTHETIC AUGMENTATION LOOP

We simulate the synthetic augmentation loop using the same deep generative models and experimental conditions as in Appendix A.1. Recall that we first require training an initial model $\mathcal{G}^1$ with a fully real dataset of $n_r^1$ samples. All subsequent models $(\mathcal{G}^t)_{t=2}^{\infty}$ are trained using $n_s^t$ synthetic samples from the previous model(s) and all of the original $n_r^1$ samples used to train $\mathcal{G}^1$. Note that each synthetic sample is always produced with sampling bias $\lambda$. Our experiments are organized as follows:

- **Generative adversarial network**: We use an unconditional StyleGAN2 architecture Karras et al. (2020) trained on the FFHQ-128×128 dataset Karras et al. (2019b). Like the StyleGAN experiment in Appendix A.1, at each generation $t \geq 2$ we sample $70k$ images with no sampling bias ($\lambda = 1$) from the immediately preceding model $\mathcal{G}^{t-1}$. However, now the synthetic dataset $\mathcal{D}_s^t$ includes samples from *all* the previously models $(\mathcal{G}^{\tau})_{\tau=1}^{t-1}$, producing a synthetic data pool of size $n_s^t = (t-1)70k$ that grows linearly with respect to $t$. The real FFHQ dataset is always present at every generation: $\mathcal{D}_r^1 = \mathcal{D}_r^t$ and $n_r^1 = n_r^t = 70k$ for every generation $t$.

- **Diffusion model**: We use a conditional MNIST-DDPM Ho et al. (2020) with $T = 500$ diffusion time steps. In this experiment the synthetic dataset $\mathcal{D}_s^t$ is only sampled from the previous generation $\mathcal{G}^{t-1}$ with sampling bias $\lambda$, and $n_r^1 = n_s^t = 60k$ for all $t \geq 2$. The original real MNIST dataset is also available at every generation: $\mathcal{D}_r^1 = \mathcal{D}_r^t$ and $n_r^1 = n_r^t = 60k$ for all $t$.

## A.3  THE FRESH DATA LOOP

As in previous autophagous loop variants, we assume that all models are initially trained solely on real samples, with the number of real samples denoted here as $n_r^1 = n_{ini}$. In subsequent generations (i.e., for $t \geq 2$) the generative models are trained with a fixed number of real samples, denoted as $n_r^t = n_r$, and a fixed number of synthetic samples, denoted by $n_s^t = n_s$. In the fresh data loop, the dataset $\mathcal{D}_r^t$ is independently sampled from the reference probability distribution $\mathcal{P}_r$, while the dataset $\mathcal{D}_s^t$ is sampled exclusively from the previous generation $\mathcal{G}^{t-1}$, with a sampling bias represented as $\lambda$.

We simulate the fresh data loop using different values for $n_{ini}, n_r, n_s,$ and $\lambda$. The Gaussian example enables examination of the fresh data loop in greater detail, especially in the asymptotic regime. Meanwhile, our MNIST-DDPM example demonstrates the impact of fresh data loop on more realistic dataset and model.

- **Gaussian model**: We consider a normal reference distribution $\mathcal{P}_r = \mathcal{N}(\mathbf{0}_d, \boldsymbol{I}_d)$ with a dimension of $d = 100$. For modeling the Gaussian distribution, we utilize an unbiased moment estimation approach, as described in Equation (1).

- **Diffusion model**: We use a conditional DDPM Ho et al. (2020) with $T = 500$ diffusion time steps. We consider the MNIST dataset as our reference distribution.

## A.4   METRICS FOR MADNESS

Ascertaining whether an autophagous loop has gone MAD or not (recall Definition 2) requires that we measure how far the synthesized data distribution $\mathcal{G}^t$ has drifted from the true data distribution $\mathcal{P}_r$ over the generations $t$. We use the notion of the Wasserstein distance as implemented by the Fréchet Inception Distance (FID) for this purpose. We will also find the standard concepts of precision and recall useful for making rigorous the notions of quality and diversity, respectively.

**Wasserstein distance**, or earth mover's or optimal transport distance (Kantorovich, 1960), measures the minimum work required to move the probability mass of one distribution to another. Computing the Wasserstein distance between two datasets (e.g., real and synthetic images) is prohibitively expensive. As such, standard practice employs the FID (Heusel et al., 2017) as an approximation, which calculates the Wasserstein-2 distance between Inception feature distributions of real and synthetic images. For our MNIST experiments we calculate FIDs using the features from a LeNet (Lecun et al., 1998) rather than an Inception network, because numerical digits are not exactly natural images.

**Precision** quantifies the portion of synthesized samples that are deemed high *quality* or visually appealing. We use precision as an indicator of sample quality. We compute precision by calculating the fraction of synthetic samples that are closer to a real data example than to their $k$-th nearest neighbor (Kynkäänniemi et al., 2019). We use the default $k = 5$ in all experiments.

**Recall** estimates the fraction of samples in a reference distribution that are inside the support of the distribution learned by a generative model. High recall scores suggest that the generative model captures a large portion of *diverse* samples from the reference distribution. We compute recall in a manner similar to precision (Kynkäänniemi et al., 2019). Given a set of synthetic samples from the generative model, we calculate the fraction of real data samples that are closer to any synthetic sample than its $k$-th nearest neighbor. In Appendix C.1 we demonstrate how recall captures synthetic diversity in an autophagous loop more accurately than variance.

## B   PROOF OF SYNTHETIC GAUSSIAN MARTINGALE VARIANCE COLLAPSE

We now prove that for the process described in Equation (1), $\boldsymbol{\Sigma}_t \xrightarrow{\text{a.s.}} \mathbf{0}$.

*Proof.* First write $\mathbf{x}_t^i = \sqrt{\lambda}\boldsymbol{\Sigma}_{t-1}^{1/2}\mathbf{z}_t^i + \boldsymbol{\mu}_{t-1}$ for $\mathbf{z}_t^i \sim \mathcal{N}(\mathbf{0}_d, \boldsymbol{I}_d)$. Then consider the process $\text{tr}[\boldsymbol{\Sigma}_t]$, which is a lower bounded submartingale:

$$\text{tr}[\boldsymbol{\Sigma}_t] = \lambda\text{tr}\left[\boldsymbol{\Sigma}_{t-1}^{1/2}\left(\frac{1}{N-1}\sum_{i=1}^{N}(\mathbf{z}_t^i - \boldsymbol{\mu}_t^{\mathbf{z}})(\mathbf{z}_t^i - \boldsymbol{\mu}_t^{\mathbf{z}})^\top\right)\boldsymbol{\Sigma}_{t-1}^{1/2}\right], \tag{4}$$

where $\boldsymbol{\mu}_t^{\mathbf{z}} = \frac{1}{N}\sum_{i=1}^{N}\mathbf{z}_t^i$. By Doob's martingale convergence theorem (Williams, 1991, Ch. 11), there exists a random variable $w$ such that $\text{tr}[\boldsymbol{\Sigma}_t] \xrightarrow{\text{a.s.}} w$, and we now show that we must have $w = 0$. Without loss of generality, we can assume that $\boldsymbol{\Sigma}_{t-1}$ is diagonal, in which case it becomes clear that $\text{tr}[\boldsymbol{\Sigma}_t]$ is a generalized $\chi^2$ random variable, being a linear combination of $d$ independent $\chi^2$ random variables with $N - 1$ degrees of freedom, mixed with weights $\lambda\text{diag}(\boldsymbol{\Sigma}_{t-1})$. Therefore, we can write $\text{tr}[\boldsymbol{\Sigma}_t] = \lambda y_t\text{tr}[\boldsymbol{\Sigma}_{t-1}]$, where $y_t$ is a generalized $\chi^2$ random variable with the same degrees of freedom but with mixing weights $\text{diag}(\boldsymbol{\Sigma}_{t-1})/\text{tr}[\boldsymbol{\Sigma}_{t-1}]$, and $\mathbb{E}[y_t|\boldsymbol{\Sigma}_{t-1}] = 1$. This implies that at least one mixing weight is greater than $1/D$ for each $t$, which means that for any $0 < \epsilon < 1$, there exists $c > 0$ such that $\Pr(|y_t - 1| > \epsilon) > c$. Now consider the case $\lambda = 1$. Since $|y_t - 1| > \epsilon$ infinitely often with probability one, the only $w$ that can satisfy $\lim_{t\to\infty}\text{tr}[\boldsymbol{\Sigma}_0]\prod_{s=1}^{t}y_s = w$ is $w = 0$. For general $\lambda \leq 1$, $\text{tr}[\boldsymbol{\Sigma}_t]$ is simply the product of the process for $\lambda = 1$ and the sequence $\lambda^{t-1}$, and so the product must also converge to zero almost surely. Finally, since $\text{tr}[\boldsymbol{\Sigma}_t] \xrightarrow{\text{a.s.}} 0$, we also must have $\boldsymbol{\Sigma}_t \xrightarrow{\text{a.s.}} \mathbf{0}$, where convergence is defined with any matrix norm. $\square$

## C  ADDITIONAL EXPERIMENTS FOR THE FULLY SYNTHETIC LOOP

Here we present additional experiments for the fully synthetic loop.

### C.1  RECALL VERSUS VARIANCE: GMMS IN AN UNBIASED FULLY SYNTHETIC LOOP

We also trained 2D GMMs in an unbiased fully synthetic loop using the same 25-mode distribution as (Che et al., 2020). In Figure 11 we see that the fully synthetic loop gradually reduces the number of modes covered by the synthetic distribution. Various metrics could measure this loss in diversity, so in Figure 12 we explore how well each metric reflects the dynamics of the fully synthetic loop, finding that recall is best-equipped to measure diversity in multimodal datasets.

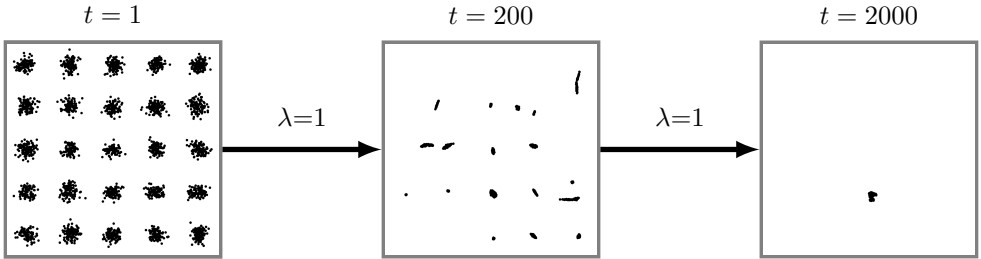

Figure 11: **The fully synthetic loop gradually causes mode collapse.** Estimated GMM (Che et al., 2020) distributions after 1, 200, and 2k iterations of an unbiased fully synthetic loop. Notice that the modes are lost asymptotically.

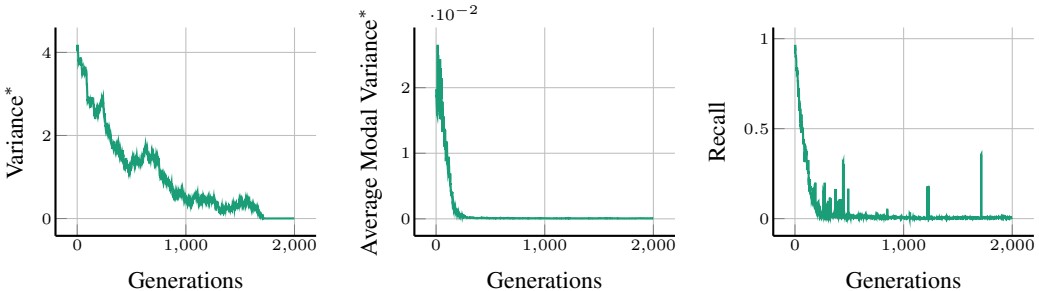

Figure 12: **Recall is the most suitable commonly accepted metric for diversity in an autophagous loop.** For GMMs in a fully synthetic loop (Figure 11), there are three primary potential metrics of diversity: variance,[7] average modal variance (the average variance of each mode), and recall (Kynkäänniemi et al., 2019). We observe that the overall variance (left) does not reflect the loss of modes that we see in Figure 11 as smoothly as recall (right) and average modal variance (middle). Recall is therefore a suitable choice for measuring diversity in multimodal datasets and, unlike average modal variance, is compatible with distributions where the number of modes is not tractable (e.g., natural images).

### C.2  WASSERSTEIN GANS IN AN UNBIASED FULLY SYNTHETIC LOOP

In this experiment we trained Wasserstein GANs (with gradient penalty) (Gulrajani et al., 2017a) on the MNIST dataset in a fully synthetic loop for 100 generations. As shown in Figure 13, the FID monotonically increases, while quality (precision) and diversity (recall) monotonically decrease.

---

[7]For multidimensional datasets, we calculate variance as the trace of covariance.

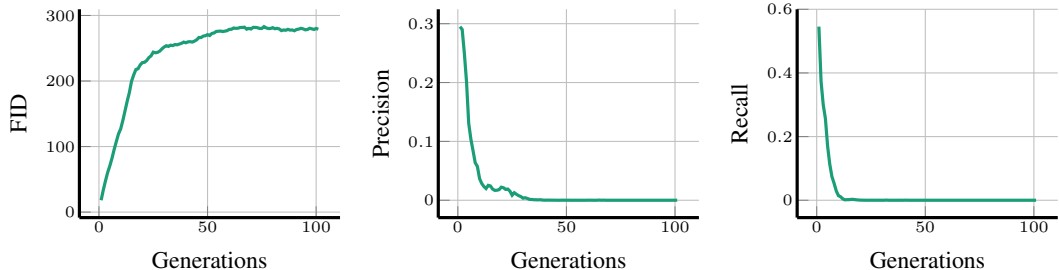

Figure 13: **The negative effects of the fully synthetic loop are monotonic and inescapable.** The FID (left), quality (precision, middle), and diversity (recall, right) of synthetic FFHQ and MNIST images produced by MNIST Wasserstein GANs.

## C.3 ADDITIONAL MNIST-DDPM FULLY SYNTHETIC LOOP RESULTS

In Figure 4 we showcased the results of training MNIST-DDPMs in a fully synthetic loop with various sampling bias factors $\lambda$. In Figure 14 we have the results (FID, precision, and recall) more generations $t$ and different sampling biases $\lambda$.

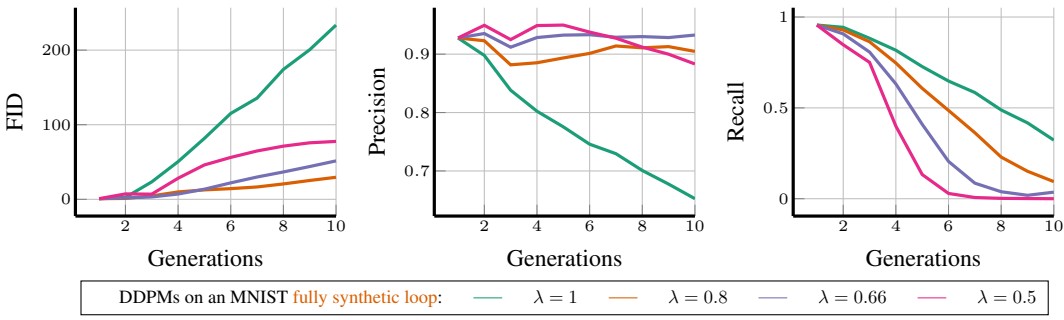

Figure 14: **In a fully synthetic loop, sampling bias $\lambda < 1$ can mitigate distributional drift and losses in quality, but only at the cost of rapid losses in diversity** The FID (left), quality (precision, middle), and diversity (recall, right) of synthetic images from an MNIST-DDPM fully synthetic loop.

## C.4 NORMALIZING FLOW FULLY SYNTHETIC LOOP

We implemented the fully synthetic loop using normalizing flows (Dinh et al., 2016; Kingma and Dhariwal, 2018) for generative modeling of the two-dimensional Rosenbrock reference distribution (Pagani et al., 2022) in order to visualize the outcome of this particular scenario in a controlled setting. Normalizing flows are unique in that they enable exact evaluation of the likelihood of the estimated distribution due to their invertibility (Dinh et al., 2016). This leads to a relatively straightforward training procedure compared to GANs, which often require careful balancing between the generator and discriminator networks to avoid mode collapse (Gulrajani et al., 2017b). Therefore, by using a low-dimensional reference distribution, this setup allows us to demonstrate the fully synthetic loop while eliminating potential training imperfections. We implemented this example using the InvertibleNetworks.jl (Orozco et al., 2023) package for normalizing flows.

According to the fully synthetic loop setup, we start with a training dataset of $10^4$ samples from the 2D Rosenbrock distribution with the density function $\mathcal{P}_r(x_1, x_2) \propto \exp\left(-\frac{1}{2}x_1^2 - \left(x_2 - x_1^2\right)^2\right)$ (Pagani et al., 2022), which is plotted on the left-hand side of Figure 15. The subsequent generations of normalizing flow models are trained using synthetic data generated by the previous pre-trained normalizing flow for 16 generations, both with and without sampling bias. We employ the GLOW normalizing flow architecture (Kingma and Dhariwal, 2018) with eight coupling layers (Kingma and Dhariwal, 2018) and a hidden dimension of 64. The training is carried out for 20 epochs with a batch size of 256 for each generation, ensuring convergence as determined by monitoring the

model's likelihood over a validation set. Figure 15 summarizes the results of this fully synthetic loop setup. To incorporate sampling bias, we sample from $\mathcal{N}(\mathbf{0}_d, \lambda \mathbf{I}_d)$ from the latent space of the model, where $d = 2$. As shown, regardless of the presence of sampling bias, the resulting distribution after 16 generations loses the tails of the reference distribution, indicating a loss of diversity. This phenomenon becomes more pronounced when sampling bias is present ($\lambda < 1$).

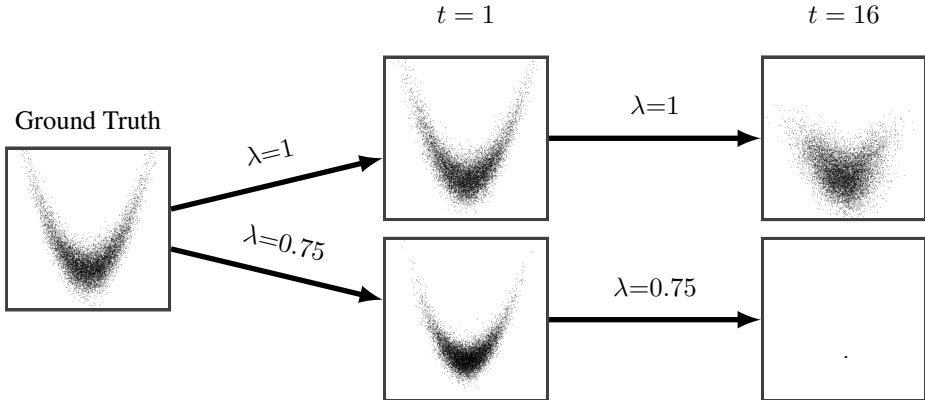

Figure 15: **Normalizing flows are not immune to the fully synthetic loop.** The fully synthetic loop implememted with a formalizing flow (Dinh et al., 2016) applied to the 2D Rosenbrock distribution (Pagani et al., 2022). Sampling with or without bias still loses the tails of the distribution (i.e., diversity). Using $\lambda < 1$ accelerates this loss of diversity.

## D FFHQ UNBIASED FULLY SYNTHETIC LOOP IMAGES

We show additional randomly chosen synthetic samples produced by the same FFHQ-StyleGAN2 unbiased fully synthetic loop as in Figure 1.

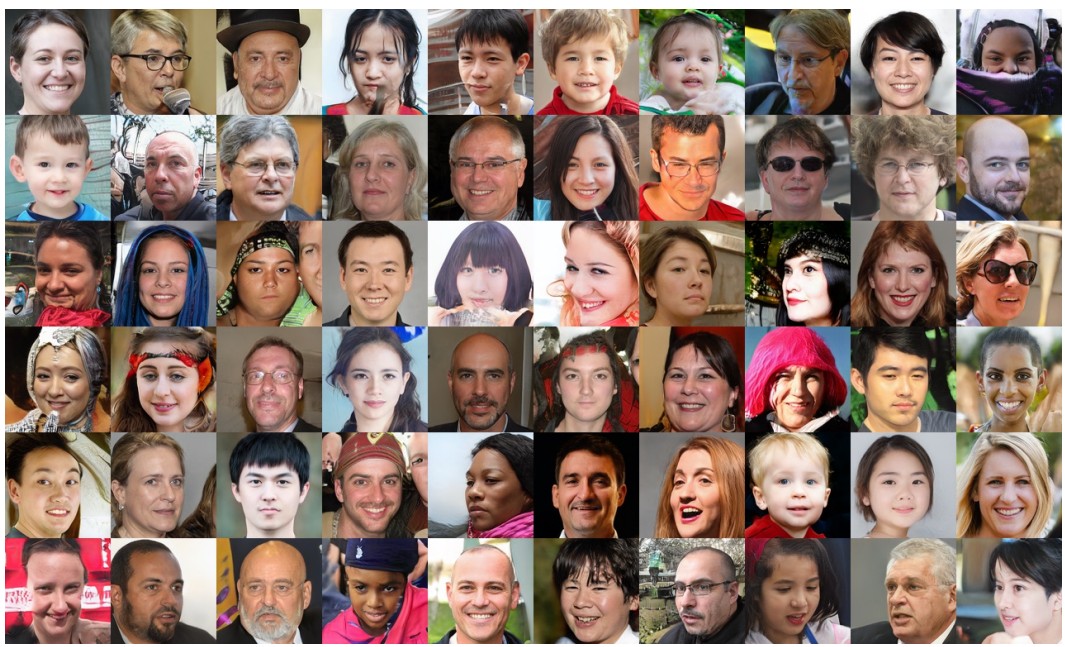

Figure 16: Generation $t = 1$ of a fully synthetic loop with bias $\lambda = 1$. i.e., synthetic samples from the first model $\mathcal{G}^1$.

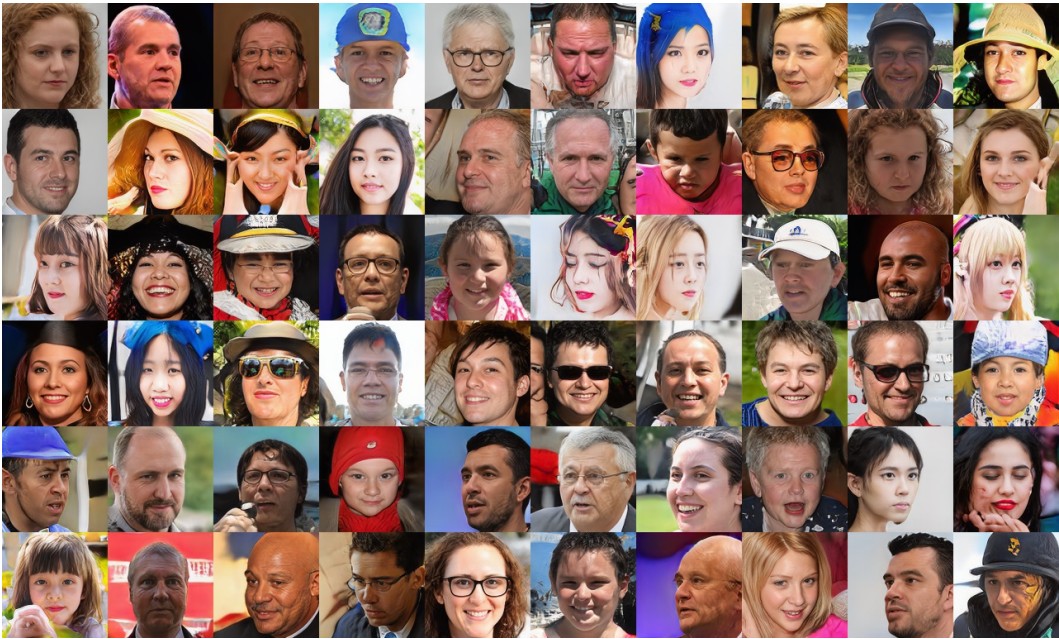

Figure 17: Generation $t = 3$ of a fully synthetic loop with bias $\lambda = 1$

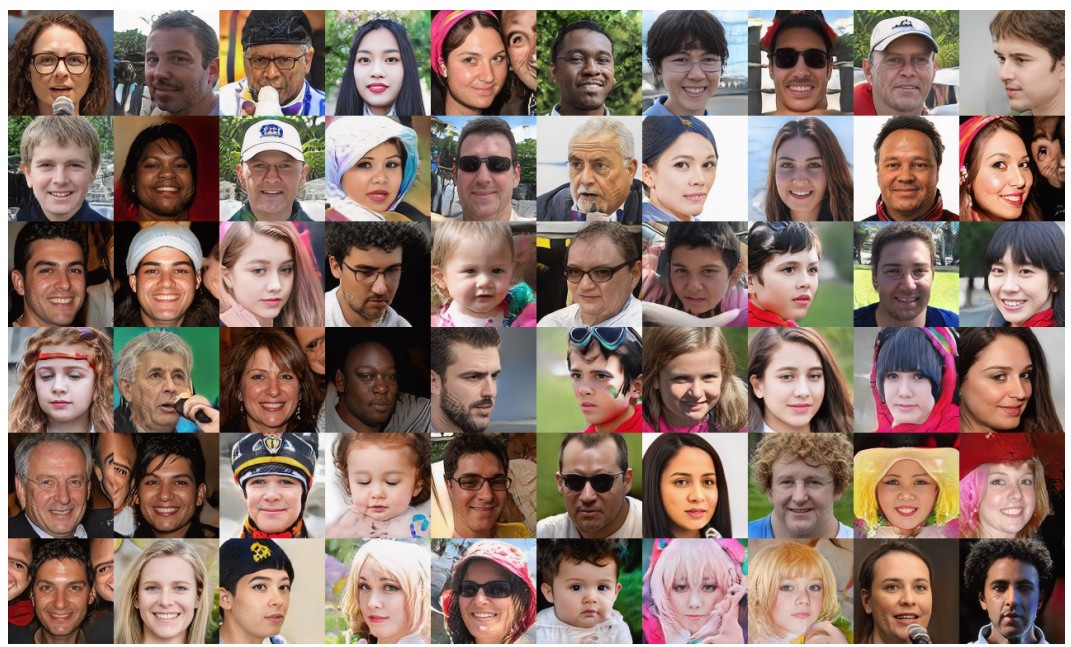

Figure 18: Generation $t = 5$ of a fully synthetic loop with bias $\lambda = 1$

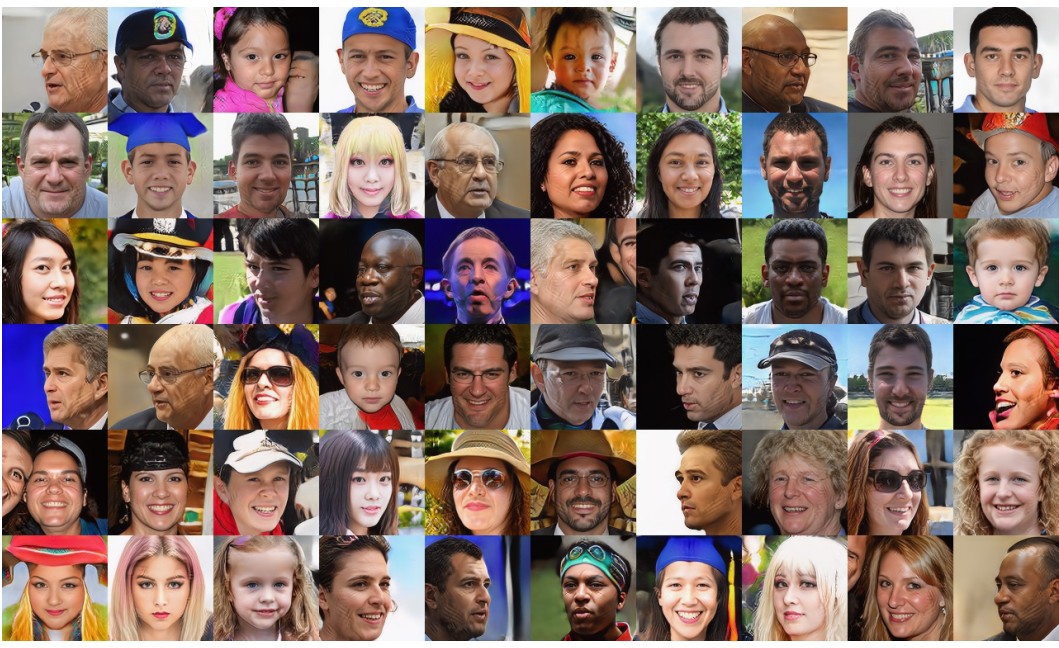

Figure 19: Generation $t = 7$ of a fully synthetic loop with bias $\lambda = 1$

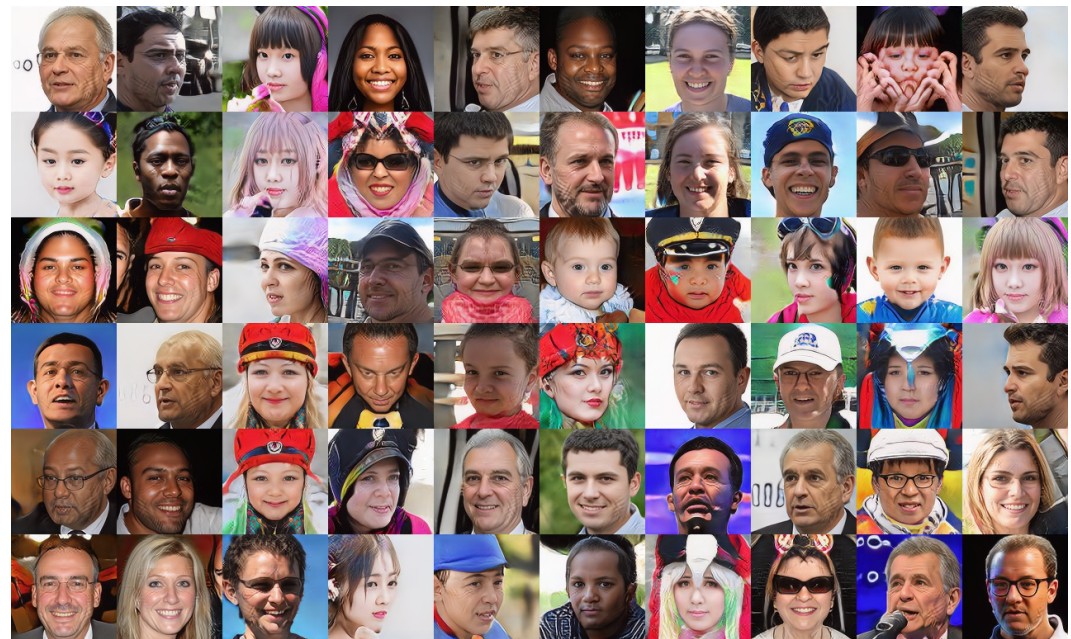

Figure 20: Generation $t = 9$ of a fully synthetic loop with bias $\lambda = 1$

# E    FFHQ BIASED FULLY SYNTHETIC LOOP IMAGES

We show additional randomly chosen synthetic samples produced by the same FFHQ-StyleGAN2 for biased ($\lambda = 0.7$) fully synthetic loop as in Figure 5.

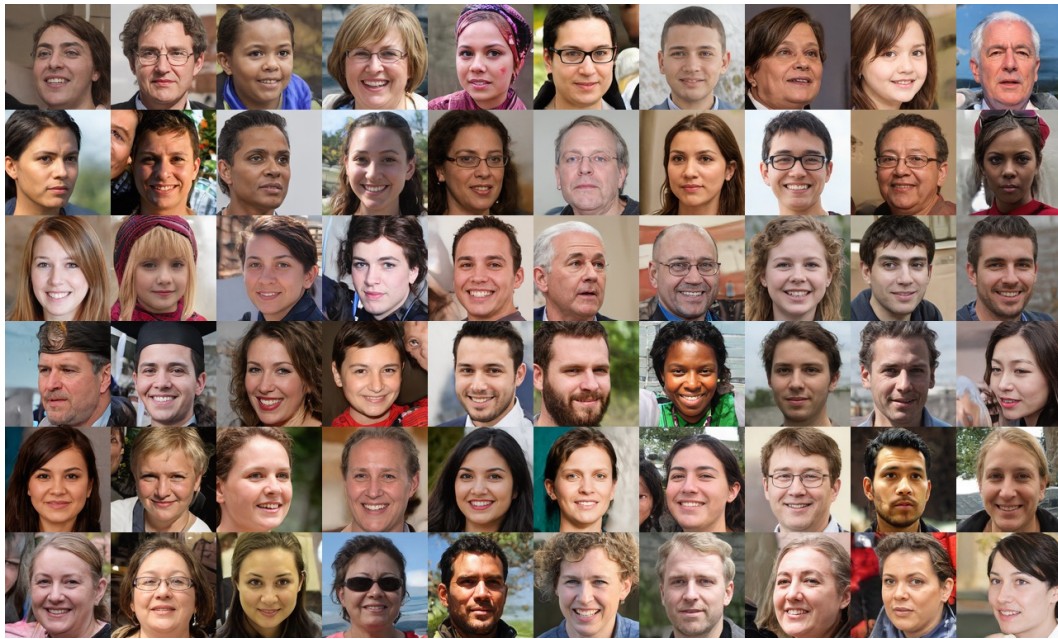

Figure 21: Generation $t = 1$ of a fully synthetic loop with bias $\lambda = 0.7$

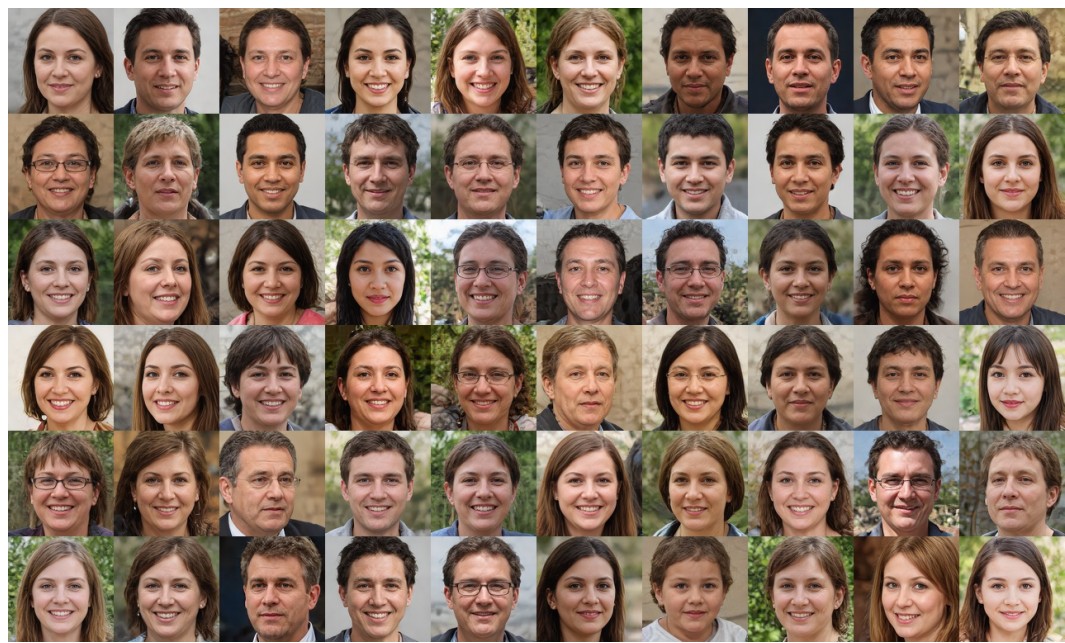

Figure 22: Generation $t = 3$ of a fully synthetic loop with bias $\lambda = 0.7$

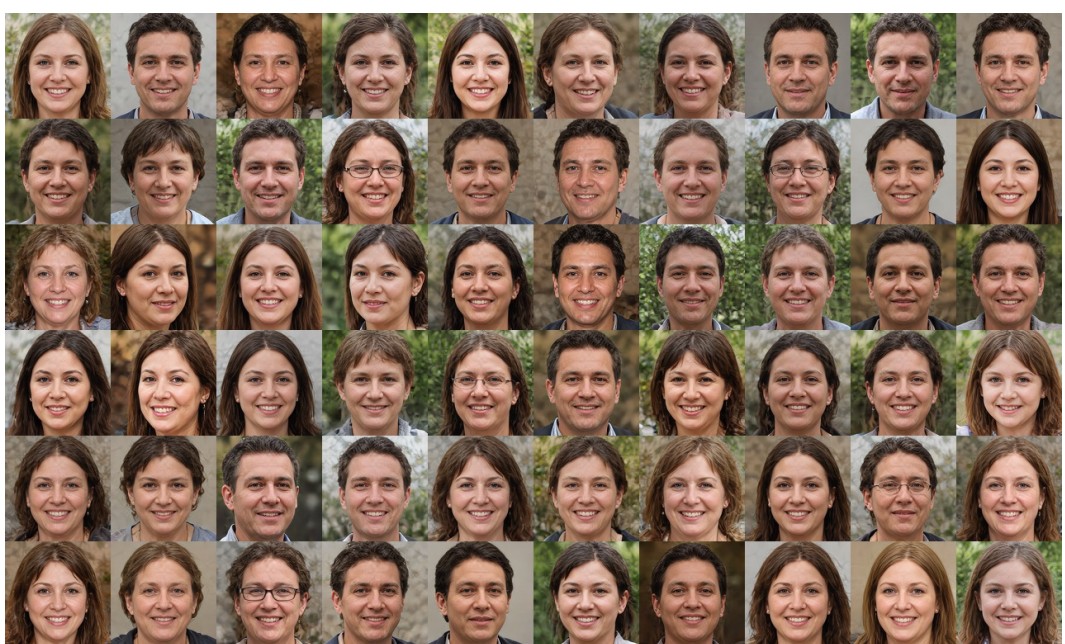

Figure 23: Generation $t = 5$ of a fully synthetic loop with bias $\lambda = 0.7$

## F MNIST-DDPM FULLY SYNTHETIC LOOP IMAGES

Here we show randomly chosen samples from each generation of an MNIST-DDPM in a fully synthetic loop for different sampling biases.

Gen

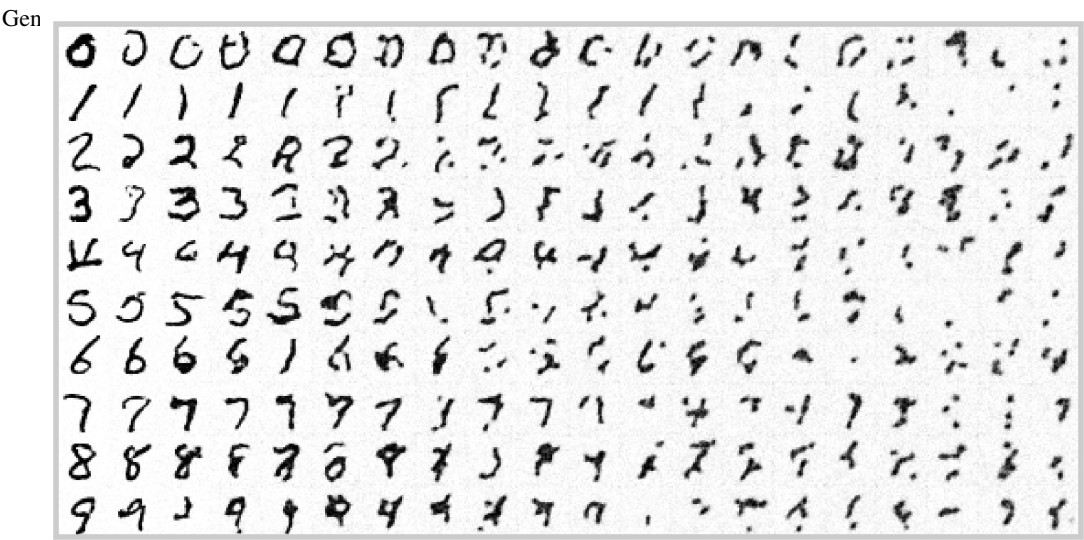

Figure 24: **Without sampling bias, synthetic data modes drift from real modes and merge together.** Randomly selected synthetic MNIST images of each generation without sampling bias ($\lambda = 1$). See Figure 6 for more details.

Gen

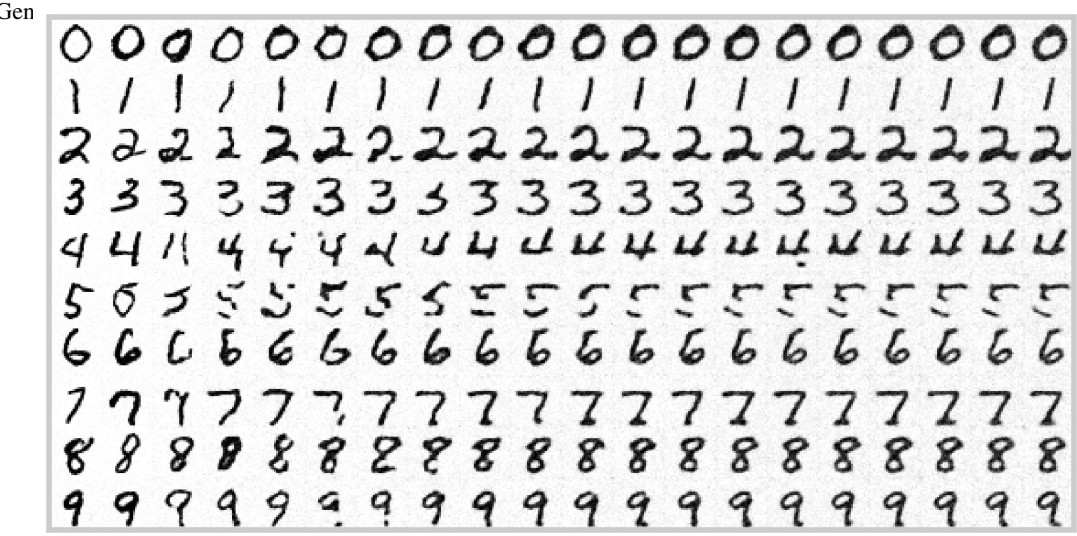

Figure 25: **With sampling bias, synthetic data modes drift and contract around just a few high-quality data points.** Randomly selected synthetic MNIST images of each generation without sampling bias ($\lambda = 0.8$). See Figure 6 for more details.

# G   FFHQ UNBIASED SYNTHETIC AUGMENTATION LOOP IMAGES

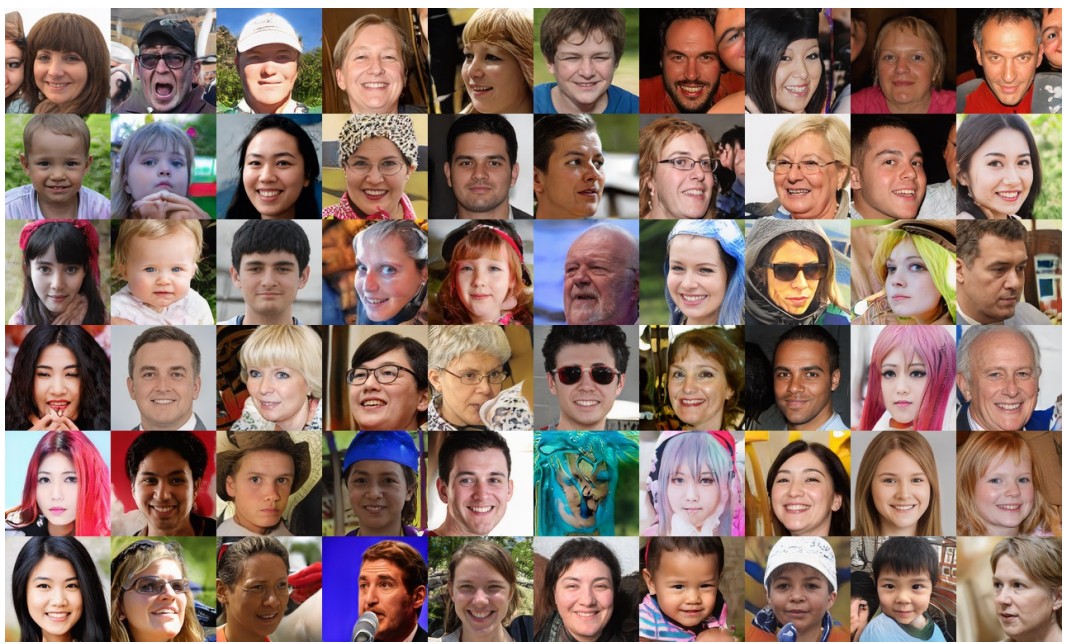

Figure 26: Generation $t = 3$ of a synthetic augmentation loop with bias $\lambda = 1$. See Figure 16 for the samples from $t = 1$ (in any autophagous loop the first model $\mathcal{G}^1$ always trains on purely real data, see Section 2).

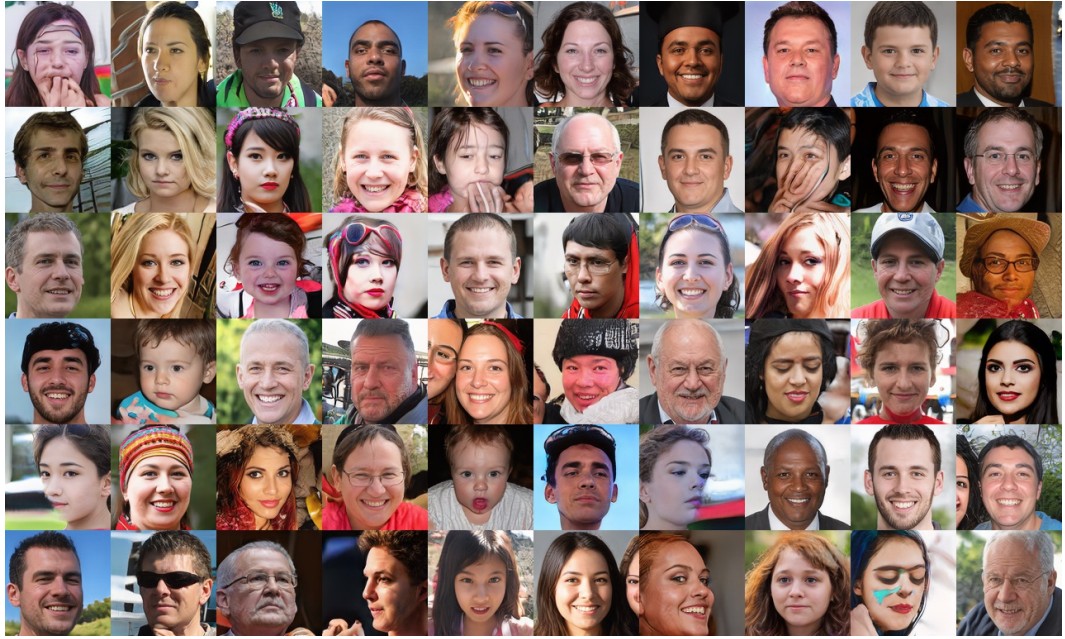

Figure 27: Generation $t = 6$ of a synthetic augmentation loop with bias $\lambda = 1$

# H ADDITIONAL RESULTS FOR THE SYNTHETIC AUGMENTATION LOOP

## H.1 THE MNIST-DDPM SYNTHETIC AUGMENTATION LOOP

In this section, we repeat the experiment in Fig 8 and described in A.2 using all synthetic data from previous generations, i.e, at each iteration $t$ we used $n_s^t = (t-1)60k$ synthetic samples from $(\mathcal{G}^\tau)_{\tau=1}^{t-1}$, combined with the initial real data we had. The results are shown in Figure 28. We see the same trend as in Figure 8. For a better comparison between using synthetic samples from all previous generations vs only using from the previous generation is shown in Figure 29. Using synthetic data from all previous generations only slows down the process the degradation in models with respect to only using data from previous generations.

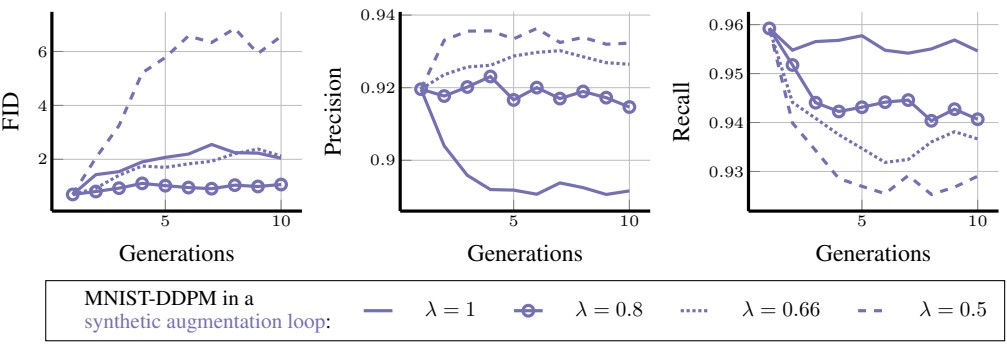

Figure 28: **Using samples from all previous generative models in synthetic augmentation loop will not stop MADness.** We show the FID, precision (quality), and recall (diversity) of MNIST-DDPM images synthesized in synthetic augmentation loops with different sampling biases $\lambda$. All three metrics exhibit the same behavior as in the case we only sample from the previous generation shown in Figure 8.

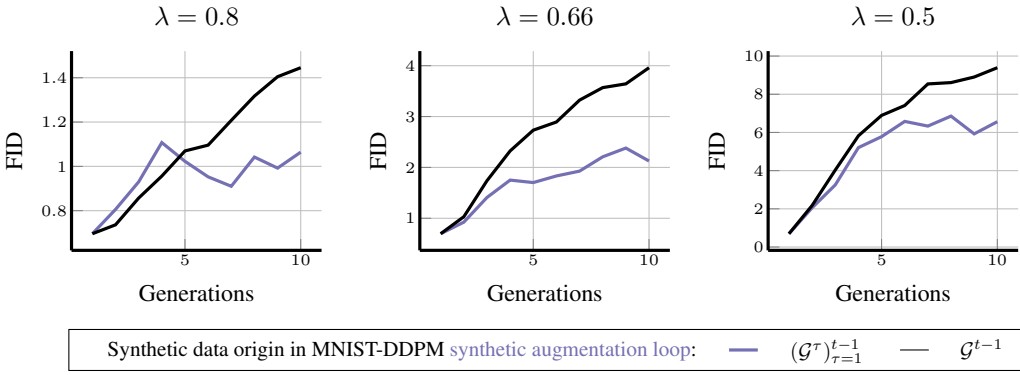

Figure 29: **Using samples from all previous generative models in synthetic augmentation loop slows the degradation of models more.** We show the FID of MNIST-DDPM images synthesized in synthetic augmentation loops with different sampling biases $\lambda$ using synthetic data from all previous generation and only the last generation.

## H.2 The Gaussian synthetic augmentation loop

In this section, we replicate the Gaussian experiment in Section 5 for synthetic augmentation loops.

In particular, we sample $n_r$ real data from the reference distribution $\mathcal{P}_r = \mathcal{N}(\mathbf{0}_d, \boldsymbol{I}_d)$ with a dimension of $d = 100$ to train the first model $\mathcal{G}^1$. For the next generations, we sample $n_s$ synthetic data from model $\mathcal{G}^{t-1}$ with sampling bias $\lambda$, and combine it with the same $n_r$ real samples we used to train $\mathcal{G}^1$. We report $\frac{n_e}{n_r}$ with $n_e$ defined in Equation 3.

We report the results for this experiment in Figure 30 and 31. We observe that for any values of $n_r > d$ with $d = 100$, the presence of synthetic samples reduces effective number of samples progressively. However, when the problem is ill-posed, i.e. $n_r < d$, $n_e$ can be increased with respect to $n_r$ if some sampling bias $\lambda$ is present in the system. However, in our experiments we observe that $n_e$ cannot surpass $n_r$ for any values of $\lambda$ or $n_e$, as it will always corresponds to an ill-posed problem $n_e < d$.

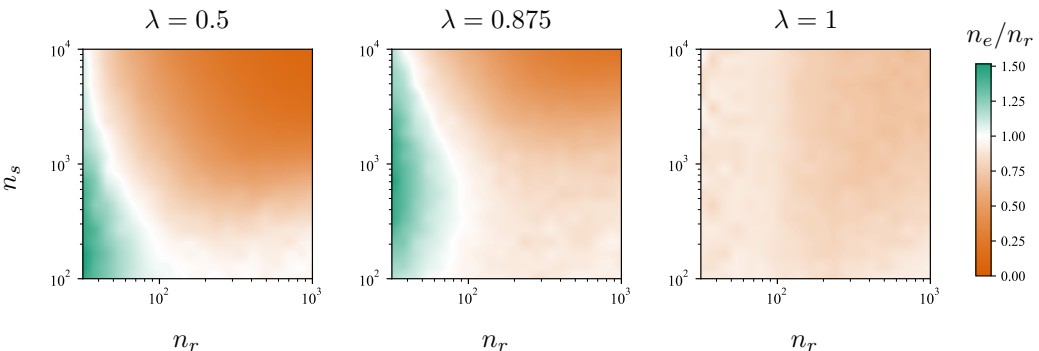

Figure 30: In a synthetic augmentation loop, we always see a decrease in $n_e$ with respect to $n_r$, except for very small values of $n_r < d = 100$, where the distribution estimation problem in ill-posed.

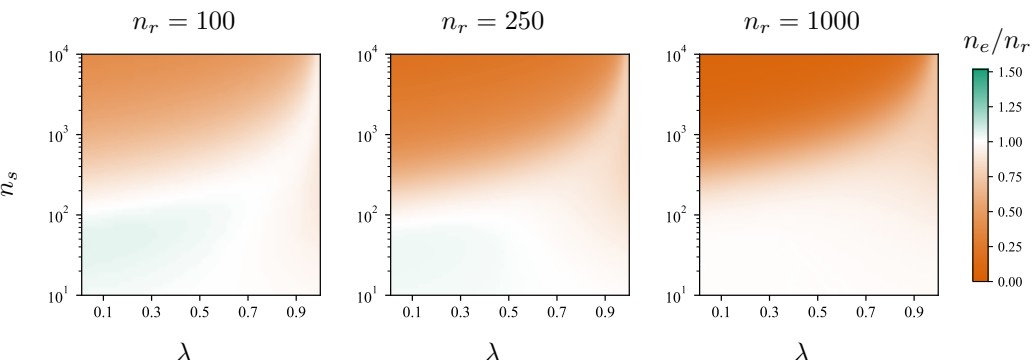

Figure 31: In a synthetic augmentation loop, we always see a decrease in $n_e$ with respect to $n_r$. Smaller values of $\lambda$ result in faster decay of $n_e$ as $n_s$ increases.

## I ADDITIONAL RESULTS FOR THE FRESH DATA LOOP

Here we provide three additional Gaussian experiments investigating the convergence of the fresh data loop.

**Experiment 1**: In Figure 10 we showed how Gaussian fresh data loop convergence depends on $n_s$ and $n_r$ for a few different values of $\lambda$. Now we depict how convergence depends on $n_s$ and $\lambda$ for a few different values of $n_r$.

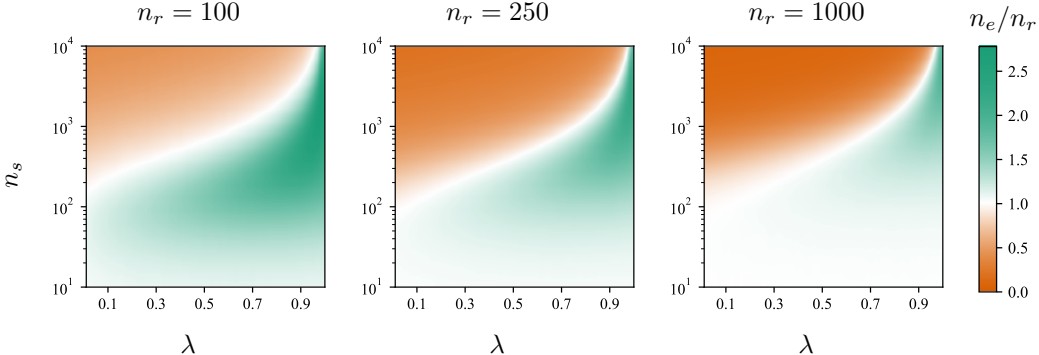

Figure 32: **In a fresh data loop, sampling bias reduces the admissible synthetic sample size.** For strong sampling bias (small $\lambda$), the maximum synthetic data count $n_s$ for which $n_e \geq n_r$ (**green** area) decreases.

**Experiment 2**: In Section 5 we assumed that we only sample from the previous generation $\mathcal{G}^{t-1}$ for creating the synthetic dataset $\mathcal{D}_s^t$. In this experiment we sample randomly from $K$ previous models $(\mathcal{G}^\tau)_{\tau=t-1-K}^{t-1}$. Here $n_r = 10^3$, $n_s = 10^4$, and $\lambda = 1$. In Figure 33 we see how $\frac{n_e}{n_r}$ varies with respect to $K$. Increasing the memory $K$ in sampling from previous generations can boost performance, however the rate of improvement becomes slower as $K$ increases. However the rate of improvement on $n_e$ is sublinear with respect to $K$.

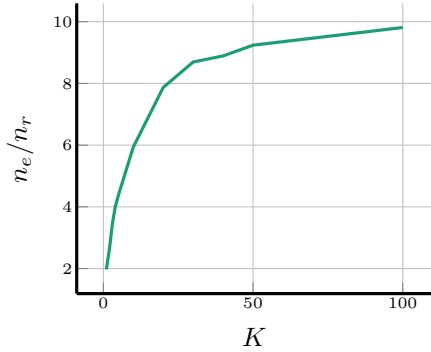

Figure 33: The effective sample size $n_e$ divided by the real sample size $n_r$ for different numbers of accessed previous generations $K$.

**Experiment 3**: Here we assume that we are sampling from an environment where $p$ percent of data is real, and the rest is synthetic data from the previous generation $\mathcal{G}^{t-1}$ with sampling bias $\lambda$. We change the total number of data in the dataset $n = |\mathcal{D}^t|$, with $n_r = p \times n$ and $n_s = (1 - n) \times p$. We show the Wasserstein distance for different $p$ and $\lambda$ in Figure 34.

Let us first examine the dynamics of the Gaussian fresh data loop without sampling bias ($\lambda = 1$). We observe in Figure 34 (left) that the Wasserstein distance (WD) decreases with respect to dataset size $n$. However, the presence of synthetic data ($p < 100\%$) decreases the rate at which the WD decreases, and increases the overall WD each generation in the fresh data loop. *This means that with presence of synthetic data in the Internet, the progress of generative models will become slower*

In the presence of sampling bias ($\lambda < 1$, Figure 34 right), we see that even for close values of $\lambda$ to 1, the Wasserstein distance follows a sub-linear trend, meaning that eventually the rate of progress in generative models will effectively stop, no matter how much (realistically) the total dataset size is increased.

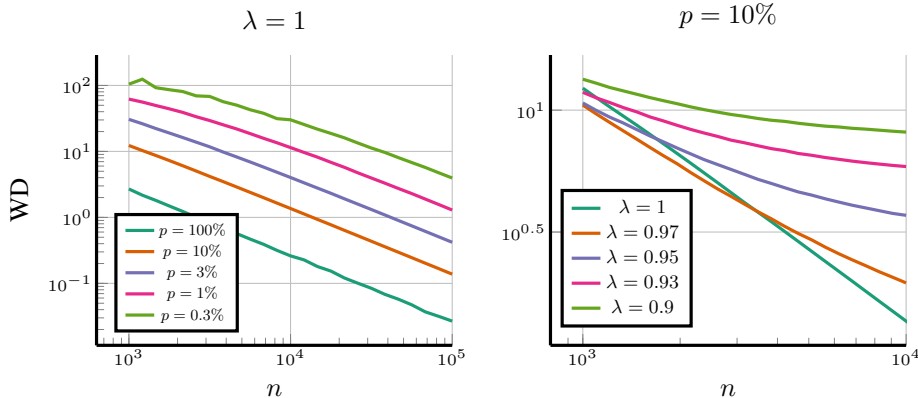

Figure 34: The Wasserstein distance (WD) versus the whole dataset size $n$, for different values of $p$ (left) and sampling bias $\lambda$ (right).

