# OpenReview forum: "Self-Consuming Generative Models Go MAD"
_ICLR.cc/2024/Conference — ICLR 2024 poster_

### Official Review · Reviewer_uRXW · 2023-10-19

**Soundness:** 3 good
**Presentation:** 3 good
**Contribution:** 3 good
**Rating:** 6
**Confidence:** 3

**Summary:**

The work investigates autophagous generative processes, where generative models train on data that includes samples from AI-synthesized data, and identifies a phenomenon termed Model Autophagy Disorder (MAD). MAD refers to the progressive deterioration of both quality and diversity in synthetic data over generations. The presence of sampling bias, common in generative model training, influences the impact of MAD. This paper shows that with enough fresh real data, the quality and diversity of the generative models do not degrade over generations. Additionally, the paper reveals a phase transition in the admissibility of synthetic data in fresh data loops, with excessive synthetic data potentially leading to MADness.  This work also discusses that autophagous loop behaviors hold across a wide range of generative models and datasets.

**Strengths:**

(1) This paper identifies and addresses a novel issue in the field of generative modeling, the phenomenon of Model Autophagy Disorder (MAD) resulting from autophagous generative processes. This issue, which has not been extensively explored before, carries potential consequences for data quality and diversity in AI systems

(2) The paper exhibits a well-structured and easily understandable presentation. It offers clear definitions, explanations, and visualizations to elucidate the concept.

(3) The study conducts extensive experiments to probe autophagous generative processes across diverse scenarios, encompassing fully synthetic loops, synthetic augmentation loops, and fresh data loops. The inclusion of various generative models, approaches, and datasets enhances the robustness of the findings.

(4) The paper's findings hold practical implications for the training of generative models, especially in situations involving synthetic data. They offer valuable insights and guidance for future model training endeavors.

**Weaknesses:**

(1) This work would benefit from a dedicated section discussing potential solutions or mitigations for Model Autophagy Disorder (MAD) in the training of generative models. This could enhance the paper's practical utility in the field.

(2) A more in-depth exploration of the broader implications of MAD is expected.  This could involve exploring the impact of MAD in various domains beyond image generation, such as NLP or audio synthesis. Additionally, discussing real-world scenarios and applications where MAD could manifest would make the paper more informative.

(3) To enhance the rigor of the paper, a more comprehensive theoretical framework should be developed to formalize the concept of MAD and its implications. A deeper exploration of the mathematical foundations underlying MAD would contribute to a stronger theoretical basis for the research.

(4) The work lacks a detailed description of the sources or methods used to generate the synthetic data used in this study. Providing specifics on how synthetic data was created for experiments would enhance transparency and reproducibility.

**Questions:**

(1) Could the authors provide more insights into potential strategies or mitigations for addressing MAD in generative model training?

(2) How might MAD impact domains beyond image generation, such as NLP or audio synthesis? Can the authors hypothesize on potential challenges and solutions in these areas?

(3) The paper discusses the impact of sampling bias (λ). Could the authors provide a more nuanced analysis of how varying levels of sampling bias affect MAD, particularly in terms of quality and diversity?

(4) Could the authors elaborate more on the mathematical foundations of MAD and its implications?

---

> ### Author Response · Authors · 2023-11-16
> **Response to the reviewer**
>
> We thank the reviewer for the thoughtful review and many suggestions for strengthening this line of research. While we believe most of the reviewer’s suggestions are either already addressed in our work or are out of the scope of this particular project, we highly value this feedback as we investigate continuations of this work. We hope that the following responses clear up what the reviewer perceived as weaknesses of our work.
>
> **This work would benefit from a dedicated section discussing potential solutions or mitigations for Model Autophagy Disorder (MAD) in the training of generative models.**
>
> We have already incorporated a discussion on this matter within the existing Discussion section, although it is limited in length due to the page restriction. We state that effort should be made toward increasing the amount of real data compared to synthetic data. As a direction for future work, we suggest watermarking techniques for synthetic data, so that the system can detect and remove them if they are unintentionally in the dataset. The question of how to address this issue is an open area of research, which our work calls the attention of the research community to.
>
> **A more in-depth exploration of the broader implications of MAD is expected. This could involve exploring the impact of MAD in various domains beyond image generation, such as NLP or audio synthesis.**
>
> We agree that adding more experiments in other data domains, such as text and audio, would bolster the general claim of the paper. However, owing to the substantial demands of experiments and the constraints of computational power, our emphasis was primarily on image data. Visual degradation is readily noticeable in this context, allowing us to explore the impacts of various settings, such as sampling bias and different autophagic loops, within a singular data modality. As we have mentioned in the Discussion and Related Work sections, the MAD effect has been observed more generally, with other works reaching similar conclusions for other modalities (such as text) in more limited settings.
>
> **To enhance the rigor of the paper, a more comprehensive theoretical framework should be developed to formalize the concept of MAD and its implications.**
>
> We are also very interested in a unifying mathematical theory of MAD. Our limited analytical results in the Gaussian case for the fully synthetic loop relied on specific properties of Gaussian parameter estimation. Developing a general theory would require additional nontrivial innovations that are outside the scope of this paper. We believe that this will be a rich line of research in the future.
>
> **The work lacks a detailed description of the sources or methods used to generate the synthetic data used in this study.**
>
> By “synthetic data” in this work, we refer to samples of a generative model that has been trained on some combination of real data and samples from previous generative models. As we stress on the first page of the paper, we do not mean synthetic data in the sense of data generated procedurally according to fixed rules or simulations, which would of course require a detailed description for proper replication. All of our results can be replicated by using the real datasets (e.g., MNIST, FFHQ) and training the corresponding sequence of generative models (e.g., DDPM, StyleGAN) following the procedures we detail in the appendices.
>
> If the reviewer still feels that we do not adequately describe the experimental process, please let us know.
>
> **Could the authors provide a more nuanced analysis of how varying levels of sampling bias affect MAD, particularly in terms of quality and diversity?**
>
> We do provide experiments on both real and Gaussian data that detail the effect of varying $\lambda$ – see Figure 14 for the fully synthetic loop, Figures 8, 28 (new), 29 (new) for the synthetic augmentation loop, and Figures 10, 30, 31, 32, 33 for the fresh data loop. We have ensured that $\lambda$ captures the trend of having value between 0 and 1 with smaller values corresponding to more sampling bias, but due to the fact that we consider many different generative models, datasets, and biasing mechanisms, we cannot ensure that $\lambda$ is calibrated in such a way that we can give any meaning to specific values. However, the general conclusion is consistent among all models and methods, where the increasing sampling bias enhances the quality of the data, at the expense of losing diversity. We recommend the existing literature for further nuance on specific sampling bias methods in different generative models.

---

> > ### Comment · Reviewer_uRXW · 2023-11-23
> >
> > Thank you for the reply. I reviewed the response and the discussions with other reviewers. I choose to maintain my current score.

---

### Official Review · Reviewer_ZeQh · 2023-10-31

**Soundness:** 3 good
**Presentation:** 4 excellent
**Contribution:** 3 good
**Rating:** 6
**Confidence:** 4

**Summary:**

This paper investigates the effect of repetition of the process that generated data of a generative model are included in the training data of generative models of next generation (self-consuming loop).
They categorize possible scenarios to 1. training data are fully generated, 2. training data are partially generated and real data are fixed, and 3. training data are partially generated and real data change.
Theoretical and empirical analyses reveal that scenarios 1 and 2 lead generative models to collapse after some iterations of the loop.

**Strengths:**

The topic, self-consuming loop, is relevant to the community, as generative models are more and more common and a plenty of generated data are released to the internet. This paper assumes three possible scenarios of the loop and analyzes each of them, which distinguishes this work.
It is also important that the authors discuss how to alleviate the effect of "MAD".

**Weaknesses:**

Although the authors define the autophagous mechanisms such that "each generative model $\mathcal{G}^t$ is trained on data that includes samples from previous models" (p3), but to my understand, the theory (around eq 1) and the experiments (described in A.1 and A.2 diffusion model) only consider the case that each model $\mathcal{G}^t$ is trained on data generated by $\mathcal{G}^{t-1}$.
The discrepancy between the definition of autophagous mechanisms and the theory and experiments is worth noting otherwise it diminishes the soundness.

**Questions:**

* To my understand, Huang et al. 2022 (cited in p3) considers "synthetic data augmentation", but Hataya et al. 2022 (cited in p3) focuses on the effects of the first loop of synthetic augmentation loops or fully synthetic loops on some downstream tasks, such as classification. Do I misunderstand something?
* Why the authors choose StyleGAN2 with FFHQ and DDPM with MNIST?

---

> ### Author Response · Authors · 2023-11-16
> **Response to the reviewer**
>
> We express our gratitude to the reviewer for providing thorough and constructive feedback on the manuscript. We have a few responses to specific points.
>
> **To my understand, the theory (around eq 1) and the experiments (described in A.1 and A.2 diffusion model) only consider the case that each model $\mathcal{G}^t$ is trained on data generated by $\mathcal{G}^{t-1}$**
>
> It is correct that the theoretical results consider training only on data generated by the previous model $\mathcal{G}^{t-1}$, but this is **not** true for all of our experiments. For example, in the synthetic augmentation loop for StyleGAN (Figure 7 in Appendix A.2) and the newly added MNIST DDPM experiment (Figures 28 and 29 in Appendix H), we accumulated synthetic data from all previous generations. As another example, in Figure 33 in Appendix I, we experiment with fresh data loops for Gaussian data using synthetic samples from all previous generations.
>
> **Huang et al. 2022 considers "synthetic data augmentation", but Hataya et al. 2022 (cited in p3) focuses on the effects of the first loop…on some downstream tasks, such as classification. Do I misunderstand something?**
>
> That is correct. Hataya et al. showed how AI-synthesized data can negatively impact downstream tasks like classification, but they did not investigate the limiting effects of autophagous loops. Huang et al. do consider synthetic data augmentation, but specifically from a perspective of improving language models in certain tasks.
>
> **Why the authors choose StyleGAN2 with FFHQ and DDPM with MNIST?**
>
> We selected these models because they achieve near-state-of-the-art performance at a reasonable computational expense. These models represent two widely used approaches to generating image data: diffusion models and GANs. Due to the substantial computational requirements for training and sampling from diffusion models, we opted for a less complex dataset (MNIST) for DDPM. In contrast, we used FFHQ for StyleGAN, as it is less computationally demanding.

---

> > ### Comment · Reviewer_ZeQh · 2023-11-22
> > **Response to the authors**
> >
> > Thank you for the reply.
> >
> > > It is correct that the theoretical results consider training only on data generated by the previous model, but this is not true for all of our experiments.
> >
> > Thank you for the clarification. It would be great if you could clearly state this in the final manuscript.
> >
> > > Due to the substantial computational requirements for training and sampling from diffusion models, we opted for a less complex dataset (MNIST) for DDPM.
> >
> > As you mentioned, MNIST is less complex and is already clustered very well. Do you think the empirical results of MNIST are applicable to more complex datasets?

---

> ### Author Response · Authors · 2023-11-23
> **2nd response to the reviewer**
>
> 1. Regarding the first comment about the manuscript's clarity concerning when some of our experiments train on multiple previous models' synthetic data rather than just the immediately preceding model's synthetic data, we hope that these existing excerpts from the manuscript will alleviate any concerns of unclarity:
>
> * Footnote 5 on Page 7 - "Unique to our StyleGAN2 synthetic augmentation loop, we linearly grow a pool of synthetic data to assess whether access to all previous generations' synthetic data could help future generations learn (see Appendix A.2).
>
> * Appendix A (Experiment Setups) Section 2 (The Synthetic Augmentation Loop) Bullet Point 1 - "Like the StyleGAN experiment in Appendix A.1, at each generation $t \geq 2$ we sample $70k$ images with no sampling bias $(\lambda = 1)$ from the immediately preceding model $\mathcal{G}^{t-1}$. However, now the synthetic dataset $\mathcal{D}^t_s$ includes samples from all the previous models $(\mathcal{G}^\tau)_{\tau=1}^{t-1}$, producing a synthetic data pool of size $n_s^t = (t-1)70k$ that grows linearly with respect to $t$. The real FFHQ dataset is always present at every generation: $\mathcal{D}_r^1 = \mathcal{D}^t_r$ and $n_r^1 = n^t_r = 70k$  for every generation $t$.
>
> * Appendix H Figure 28 - "Using samples from all previous generative models in synthetic augmentation loop will not stop MADness."
>
> * Appendix H Figure 29 - "Using samples from all previous generative models in synthetic augmentation loop slows the degradation of models more."
>
> * Appendix I Experiment 2 - "In Section 5 we assumed that we only sample from the previous generation $\mathcal{G}^{t-1}$ for creating the synthetic dataset $\mathcal{D}^t_s$. In this experiment we sample randomly from $K$ previous models $(\mathcal{G})_{\tau=t-1-K}^{t-1}$. Here $n_r = 10^3, n_s = 10^4,$ and $\lambda=1$. In Figure 33 we see how $\frac{n_e}{n_r}$ varies with respect to $K$. Increasing the memory $K$ in sampling from previous generations can boost performance, however the rate of improvement becomes slower as $K$ increases. However the rate of improvement on $n_e$ is sublinear with respect to $K$.
>
>
> 2. Regarding the second comment about the applicability of MNIST DDPM findings to other datasets: when we said "we opted for a less complex dataset (MNIST) for DDPM", we meant complex in the sense of lower-dimensionality, since training many diffusion models sequentially and with different guidances (sampling biases) would be computationally infeasible. Even training StyleGAN, which is far faster than SOTA diffusion models, takes roughly 1 week per model. Furthermore, sampling from diffusion models is quite lengthy since the quality of synthethic images generally scales with the number of forward passes applied during synthesis. Even using another low-dimensional dataset like CIFAR10 would still cause our computational costs to grow by a factor of $(32\times32\times3)/(28\times28) \approx 4$, so to efficiently investigate the effects of multi-generation autophagy on diffusion models, MNIST is the best option; this is especially true when considering the fresh data loop, which requires additional hyperparameter tuning due to the reduction in each model's training dataset size. Lastly and most importantly, given that our findings with MNIST DDPMs agree with those we see in FFHQ StyleGANs, there is no reason to doubt that our empirical results apply to higher-dimensional datasets.

---

### Official Review · Reviewer_VGNe · 2023-11-01

**Soundness:** 3 good
**Presentation:** 3 good
**Contribution:** 3 good
**Rating:** 8
**Confidence:** 3

**Summary:**

This paper investigates performance degradation of generative models in self-consuming (or autophagous) loops. The paper categorizes three scenarios of self-consuming based on real-world applications. In particular, a new scenario called "fresh data loop" is considered for the first time, where new data comes into the self-consuming loop. The empirical investigation on the scenarios reveals that, while the self-conuming loop tends to couse the degradation of the generative models, we can avoid it if the generative models in the loop continues to be exposed to new data. Also, the paper investigates how sampling techniques for better generation affects the degradation in the loop. It is empirically confirmed that these findings hold for different generative models or datasets.

**Strengths:**

- The paper is clearly written and easy to follow. Their problem is appropriately formulated in a mathematically sound way.
- In Section3, a toy example of gaussian models is analyzed, which provides a theoretical evidence for their heuristic Claim in Section 2.
- It is predictable but also insightful that the performance degradation in the loop is caused only when using fixed samples over the loop.

**Weaknesses:**

- It is unclear what Claim in Section 2 aims for. Although it is proved for a toy model of gaussians, I assume that it is aimed for a hypothesis for general case but the authors seem not to explain such aspects. For example, is random walkness in Claim observed for practical models? If not, the claim would be overclaimed and somewhat doubtful for general case. Finally note that "Claim" in a paper is usually considerred as to be proven in the paper, not a hypothesis.
- Eq. (2) is totally unclear. Is it a hypothesis or theorem? Where is $WD(n_r, n_s, \lambda)$ defined? If it is just a hypothesis, the author should make it clear in the text.
- It is unclear why $n_{ini}$ is considered in the first part of Section 5. How does it relate to the main claims in Introduction?
- The observation of a phase transition is interesting, but it suddenly appears at Section 5. I recommend to briefly discuss the motivation and the results in Introduction.

**Questions:**

See Weaknesses.

---

> ### Author Response · Authors · 2023-11-16
> **Response to the reviewer**
>
> We thank the reviewer for the detailed and constructive comments regarding the paper. We have the following remarks for your comments.
>
> **Is random walkness in Claim observed for practical models?**
>
> Thank you for pointing out this ambiguity in the “random walk” aspect of the Claim from Section 2. To clarify the claim and remove any possible connotation with a “pure” random walk (e.g., Brownian motion), we have removed “that follows a random walk” from the Claim.
>
> **Eq. (2) is totally unclear… Where is $WD(n_r, n_s, \lambda)$ defined?**
>
> $WD(n_r, n_s, \lambda)$ represents the asymptotic Wasserstein distance that models converge to in the fresh data loop, which we have observed in our experiments depends solely on $n_r, n_s,$ and $\lambda$ (the amount of new real data, new synthetic data, and sampling bias at each iteration). We have modified Eq. (2) slightly to denote that we define $WD$ as the limiting value to increase clarity.
>
> **How does [$n_{ini}$] relate to the main claims in Introduction?**,  **The observation of a phase transition is interesting…I recommend to briefly discuss the motivation and the results in Introduction.**
>
> Based on this useful feedback, we have revised the fresh data loop takeaway in the Introduction to better reflect the observations in Section 5 regarding the independence of the limiting behavior on $n_{ini}$ and regarding the phase transition in improvement due to the inclusions of synthetic data.

---

### Author Response · Authors · 2023-11-16
**Changes in the new version of the paper**

We thank the reviewers for their valuable time and feedback. We are happy to hear that the reviewers
1. Found our work interesting, timely, and novel in the context of modern generative data processes.
2. Found our presentation clearly written and our explanations and visualizations meaningful.
3. Found our examination of diverse scenarios of autophagous generative processes to be thorough and distinguished from other work in the area.

We are also very grateful for the constructive criticism. The reviewers identified a few weaknesses in our paper:
1. Reviewer VGNe raised issues of clarity in our presentation and notation.
2. Reviewer ZeQh raised an issue of clarity in our experiments, regarding whether we consider autophagous loops that train on data from more than just the previous generation.

To address these concerns, we have made the following changes in the revised version of our paper (marked in blue):
1. Per reviewer VGNe’s request, we revised the remarks on the fresh data loop in the Introduction to better convey the results in Section 5 of the paper. (page 3)
2. Per reviewer VGNe’s request, we revised the definition of autophagous loops to remove the term “random walk” to avoid conflation with Brownian motion. (page 3)
3. Per reviewer VGNe’s request, we revised the text to better clarify Eq 2. (page 8)
4. Per reviewer ZeQh’s comment, we revised the text in Appendix A.2 to emphasize clearly that we use samples from all previous generations in the StyleGAN2 FFHQ experiment in the synthetic augmentation loop. (page 14)
5. Per reviewer ZeQh’s comment, we added a new set of experiments in a new section in Appendix H.1 for the DDPM MNIST synthetic augmentation loop where we use samples from all previous generations. (page 25)

---

### Meta-Review · Area_Chair_zp91 · 2023-12-09

**Metareview:**

The authors study the problem of training generative models on synthetic data generated from prior versions of the same model. This is a timely problem due to the proliferation of generated data on the web, which is likely to contaminate future datasets used to train large generate AI models. The authors perform both a theoretical analysis of this problem on a toy dataset as well as an empirical analysis of this problem on real generative models. They study whether performance degrades under various configurations of real vs. synthetic data, leading to interesting insights about when the performance degradation arises.

The reviewers generally agreed that the paper is studying an important problem. There were concerns about the limitations of the theoretical analysis, which is on a very toy version of the problem, making it unclear how closely connected it is to realistic problems. There were also some concerns about the clarity of the writing, which the authors have worked on addressing. Finally, the paper focuses on visual domains, which limits the generality of the findings. The text domain is particularly relevant given the proliferation of AI generated text/code, and can be qualitatively different from the visual domain due to its discrete nature. While evaluating on state-of-the-art language models is challenging, the authors might consider evaluations on smaller language models to improve the strength of their results.

**Justification For Why Not Higher Score:**

While the paper studies a compelling problem, its implications are somewhat limited due to its focus on the visual domain.

**Justification For Why Not Lower Score:**

The paper is studying a compelling problem using an interesting theoretical an empirical analysis.

---

### Decision · Program_Chairs · 2024-01-16

Accept (poster)